# FROM EXPLORATION TO MASTERY: ENABLING LLMS TO MASTER TOOLS VIA SELF-DRIVEN INTERACTIONS

**Changle Qu**[1], **Sunhao Dai**[1], **Xiaochi Wei**[2], **Hengyi Cai**[2],
**Shuaiqiang Wang**[2], **Dawei Yin**[2], **Jun Xu**[1]*, **Ji-Rong Wen**[1]
[1]Gaoling School of Artificial Intelligence, Renmin University of China, [2]Baidu Inc.
{changlequ,sunhaodai,junxu,jrwen}@ruc.edu.cn, weixiaochi@baidu.com
caihengyi@ict.ac.cn, shqiang.wang@gmail.com, yindawei@acm.org

## ABSTRACT

Tool learning enables Large Language Models (LLMs) to interact with external environments by invoking tools, serving as an effective strategy to mitigate the limitations inherent in their pre-training data. In this process, tool documentation plays a crucial role by providing usage instructions for LLMs, thereby facilitating effective tool utilization. This paper concentrates on the critical challenge of bridging the comprehension gap between LLMs and external tools due to the inadequacies and inaccuracies inherent in existing human-centric tool documentation. We propose a novel framework, **DRAFT**, aimed at **D**ynamically **R**efining tool documentation through the **A**nalysis of **F**eedback and **T**rials emanating from LLMs' interactions with external tools. This methodology pivots on an innovative trial-and-error approach, consisting of three distinct learning phases: experience gathering, learning from experience, and documentation rewriting, to iteratively enhance the tool documentation. This process is further optimized by implementing a diversity-promoting exploration strategy to ensure explorative diversity and a tool-adaptive termination mechanism to prevent overfitting while enhancing efficiency. Extensive experiments on multiple datasets demonstrate that DRAFT's iterative, feedback-based refinement significantly ameliorates documentation quality, fostering a deeper comprehension and more effective utilization of tools by LLMs. Notably, our analysis reveals that the tool documentation refined via our approach demonstrates robust cross-model generalization capabilities.

## 1 INTRODUCTION

Tool learning (Mialon et al., 2023; Qin et al., 2023b; Schick et al., 2024; Qu et al., 2024b), which integrates external tools with large language models (LLMs), has significantly enhanced the capability of LLMs to address complex real-world tasks (Nakano et al., 2021; Qin et al., 2023a; M. Bran et al., 2024). By leveraging external tools, LLMs are able to mitigate the limitations of outdated pre-training data and the text-in-text-out interface, enabling them to access up-to-date information, interact with dynamic environments, and take actions beyond their original scope (Zhuang et al., 2024; Wang et al., 2024a). To effectively utilize these external tools, LLMs are typically provided with tool documentation as context (Shen et al., 2024; Song et al., 2023; Xu et al., 2023). This documentation provides essential information on how tools function, their potential uses, and the ways in which they can be leveraged to solve complex tasks. By incorporating tool documentation within the task instructions, LLMs can leverage their in-context learning abilities to understand and utilize the tools efficiently (Wei et al., 2022; Hsieh et al., 2023). Therefore, tool documentation is an indispensable component driving the success of tool learning, serving as a bridge between LLMs and external tools.

However, existing tools primarily originate from pre-established, human-engineered code repositories and are not explicitly tailored for the utilization of LLMs from their inception, let alone the corresponding tool documentation. In fact, orchestrating an ideal documentation for an external tool that adapts to the specific requirements of LLMs remains a challenging endeavor. **First**, the original human-crafted tool documentation is typically created with human intuition in mind, and

---

* Jun Xu is the corresponding author.

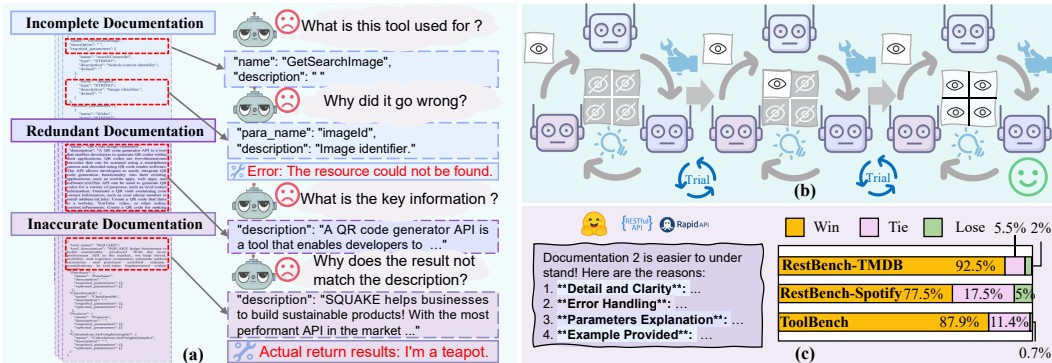

Figure 1: (a): An illustration showcasing issues in current tool documentation, including incomplete, redundant, and inaccurate information. (b): An illustration of how LLMs incrementally enhance their understanding of tool usage through iterative exploration, transitioning from untested functionalities (gray) to discovered capabilities (white) via a trial-and-error process. (c): The comparison between the raw and improved tool documentation regarding its usefulness on three tool learning datasets, highlighting that DRAFT is able to produce tool documentation that is more favored by LLMs.

is often fraught with inconsistency and ambiguities, as it primarily caters to human understanding and usually lacks the precision required for machine interpretation (Chemero, 2023; Yuan et al., 2024). As illustrated in Figure 1 (a), incomplete documentation makes it difficult for LLMs to clearly understand the purpose of a tool and when it should be invoked, while redundant documentation, containing excessive irrelevant information, obscures key details and increases token consumption in prompts. Additionally, inaccurate documentation that does not reflect the tool's actual capabilities can lead to discrepancies between the tool's outputs and its description, leading to the potential misuse of the tool by LLMs. These issues obstruct the effective utilization of tools by LLMs. **Second**, manual modification of these documentations, even with meticulous revisions, struggle to fully encompass all aspects of tool usage, since discerning the specific scope a tool can manage and identifying its edge use-cases often necessitates considerable hands-on experience. For example, as depicted in Figure 1 (a), incomplete documentation also fails to mention certain constraints on the parameters, and the LLM, being unaware of this undocumented constraint, generates an error when invoking the tool with an invalid parameter value. Addressing these issues by manually correcting or enhancing tool documentation is time-consuming and labor-intensive. Moreover, this may not scale to a substantial number of tools effectively. **Furthermore**, the dynamic nature of tool development further exacerbates this issue, as functionalities of tools are frequently updated, deprecated, or extended. Maintaining an up-to-date and accurate representation of such evolving functionalities within the tool documentation becomes an arduous task. This misalignment between the tool documentation with the current state of the tool hinders the efficient and correct utilization of tools by LLMs.

Humans, on the contrary, acquire tool proficiency through repeated interactions and hands-on experiences, capable of maintaining an updated comprehension of these tools despite their evolving functionalities. In light of this, this paper proposes DRAFT, conceptualized to automate the adjustment and optimization of tool documentation based on the feedback derived from the LLM's interaction with the tool, aiming to bridge the comprehension gap between LLMs and external tools.

More concretely, DRAFT implements a trial-and-error methodology to incrementally improve the tool documentation. As shown in Figure 2, DRAFT orchestrates three dynamically interlinked phases, which collectively facilitate the iterative process of documentation enhancement. It first undertakes the simulation of potential tool application scenarios, crafting explorative instances and capturing tool execution outcomes through a designed explorer. Subsequently, the analyzer dissects the prevailing documentation, amalgamating insights from the explorer's findings and feedback to moot documentation modification propositions. Finally, the rewriter amalgamates these insights, refining the tool documentation while simultaneously guiding further explorative pursuits by the explorer. To optimize this process, we design a diversity-promoting exploration strategy to ensure diversity in exploration, thus providing a wider range of samples for subsequent rewriting. Recognizing that different tools vary in complexity for LLMs, we introduce a tool-adaptive termination mechanism to improve efficiency during modifications by halting the iterative process once the documentation aligns with the comprehension of LLMs, thereby saving time and resources while preventing overfitting.

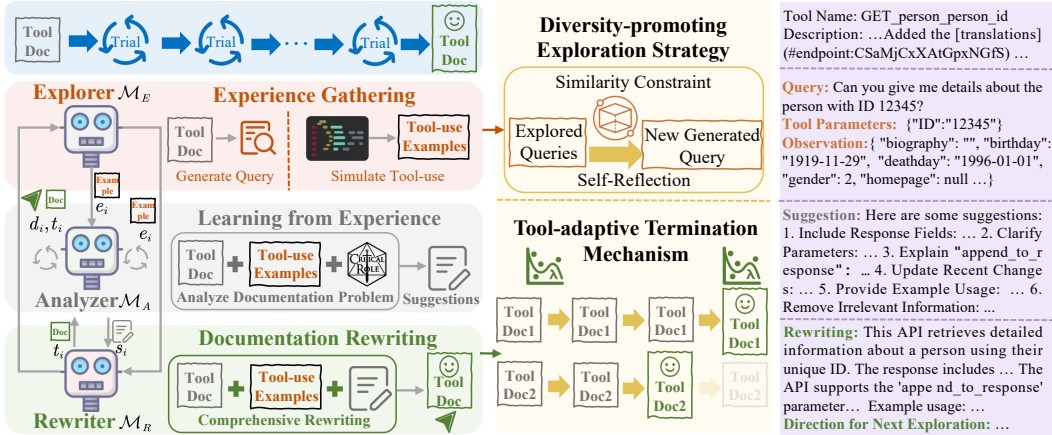

Figure 2: Schematic illustration of our proposed self-driven iterative improvement framework, DRAFT. The **left** part illustrates the three distinct learning phases in our framework, where a trial refers to a single iteration of the process. The **middle** part shows the two specialized mechanisms designed to optimize the process. The **right** part depicts an example of using DRAFT to iteratively modify the tool documentation (a complete revision trajectory is presented in Appendix E).

Through this collaborative, iterative framework, the documentation progressively morphs, aligning more coherently with LLMs operational requisites, and thus empowering LLMs to leverage external tools more effectively in their problem-solving endeavors.

In summary, our contributions are as follows: (1) We highlight that due to the inherent understanding gap between LLMs and humans, inefficiencies and inaccuracies within existing tool documentation hamper the effective utilization of tools by LLMs. (2) We introduce a novel framework, DRAFT, designed to dynamically adjust and optimize tool documentation based on the interaction feedback between LLMs and external tools, which significantly bridges the gap between them by enabling the LLMs to better comprehend and utilize the tools at their disposal, thereby enhancing the overall tool-using capabilities of LLMs. (3) Extensive experiments demonstrate that DRAFT significantly improves the quality of tool documentation and enhances the ability of LLMs to utilize external tools.

## 2 METHODS

In this section, we will first introduce the overview of DRAFT, and then provide detailed explanations of the three included learning stages. The learning algorithm is presented in Algorithm 1.

### 2.1 OVERVIEW OF DRAFT

To address the challenges of inadequate, ambiguous, and outdated tool documentation that hinder LLMs from effectively utilizing external tools, we propose DRAFT, a framework that iteratively refines tool documentation to bridge the comprehension gap between LLMs and tools. As illustrated in Figure 2, DRAFT operates through three interconnected phases: experience gathering, learning from experience, and documentation rewriting. In the experience gathering phase, an explorer simulates diverse tool usage scenarios, collecting data on how the LLM interacts with the tool based on the current documentation, thus uncovering misunderstandings and limitations. The learning from experience phase involves an analyzer examining this data to identify discrepancies between intended and actual tool usage, pinpoint ambiguities or inaccuracies in the documentation, and propose targeted improvements. In the documentation rewriting phase, a rewriter integrates these insights to update the documentation, enhancing clarity and alignment with the tool's functionalities. Through this trial-and-error framework, DRAFT is capable of simulating the process by which humans acquire proficiency in tool usage through repeated interactions and hands-on experiences, thereby automating the creation of tool documentation specifically designed for LLMs. Furthermore, by employing the diversity-promoting exploration strategy and tool-adaptive termination mechanism, DRAFT efficiently converges on optimized documentation, enabling LLMs to utilize external tools more effectively despite the initial documentation shortcomings.

---

**Algorithm 1:** The Learning Algorithm of DRAFT

---

**Input:** Raw tool documentation set $\mathcal{D}$, iteration round $I$, similarity threshold $\phi$, termination threshold $\tau$.
**Output:** Revised tool documentation set $\tilde{\mathcal{D}}$.

1 Initialize the revised tool documentation set $\tilde{\mathcal{D}} \leftarrow \emptyset$
2 **for** *raw tool documentation* $t \in \mathcal{D}$ **do**
3      **for** $i = 1$ *to* $I$ **do**
4          `// Experience Gathering` ($\S$ 2.2)
5          Instruct `Explorer` to generate an exploratory instance $e_i$ using Eq. (1)
6          **while** $\max_{j<i} \text{sim}(\mathbf{e}_i^q, \mathbf{e}_j^q) > \phi$ **do**
7              Instruct `Explorer` to generate a new exploratory instance $e_i$
8          **end**
9          Instruct `Explorer` to capture the outcomes of tool execution $r_i$
10          `// Learning from Experience` ($\S$ 2.3)
11          Instruct `Analyzer` to learn from experience and provide suggestions $s_i$ for modifications using Eq. (3)
12          `// Documentation Rewriting` ($\S$ 2.4)
13          Instruct `Rewriter` to revise the documentation based on experience and suggestions to get revised tool documentation $t_i$ and propose new exploration directions $d_i$ using Eq. (4)
14          Calculate the similarities $\Delta$ between $t_{i-1}$ and $t_i$ using Eq. (5)
15          **if** $\Delta > \tau$ **then**
16              Break
17          **end**
18      **end**
19      Updating the revised tool documentation set $\tilde{\mathcal{D}} \leftarrow \tilde{\mathcal{D}} \cup t_i$
20 **end**
21 **return** $\tilde{\mathcal{D}}$

---

## 2.2 EXPERIENCE GATHERING

In the experience gathering phase, we design an `Explorer` $\mathcal{M}_E$ to simulate plausible scenarios in which the tool may be utilized. This approach parallels the manner in which individuals investigate the potential applications of a new tool when they are unable to comprehend the accompanying manual.

Specifically, at the $i$-th iteration, the `Explorer` $\mathcal{M}_E$ generates an exploration instance $e_i$ based on the current tool documentation $t_{i-1}$, next-step exploration direction $d_{i-1}$ from the `Rewriter` $\mathcal{M}_R$, and the previous history $\mathcal{H}_i = \{(e_j, r_j) | j < i\}$, which consists of prior exploration instances $e_{<i}$ and their corresponding return results of the tool $r_{<i}$. This process is formalized as follows:

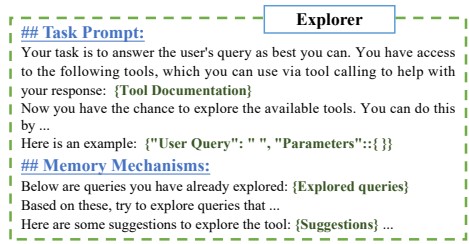

**Explorer**

**## Task Prompt:**
Your task is to answer the user's query as best you can. You have access to the following tools, which you can use via tool calling to help with your response: **{Tool Documentation}**
Now you have the chance to explore the available tools. You can do this by ...
Here is an example: **{"User Query": " ", "Parameters"::{ }}**
**## Memory Mechanisms:**
Below are queries you have already explored: **{Explored queries}**
Based on these, try to explore queries that ...
Here are some suggestions to explore the tool: **{Suggestions}** ...

Figure 3: Prompt template for explorer.

$$e_i = \mathcal{M}_E(t_{i-1}, d_{i-1}, \mathcal{H}_i), \tag{1}$$

where $e_i$ consists of a user query $e_i^q$ related to the tool and the necessary parameters $e_i^p$. The initial tool documentation is denoted as $t_0$, representing the raw documentation provided in the dataset. After generating $e_i$, the `Explorer` invokes the tool to obtain the result $r_i$ returned by the tool.

Given that tool utilization often involves complex parameter ranges, combinations, and potential error sources, it is crucial to ensure diversity in the exploration phase to cover a wide spectrum of possible scenarios (Hong et al., 2018; Friedrich et al., 2009). To address this, besides maintaining a record of all previously explored queries—which we provide to the `Explorer` to instruct it to generate instances that differ from those already generated—we also implement a **diversity-promoting exploration strategy**:

**Similarity Constraint.** When generating a new instance, the `Explorer` calculates the cosine similarity between the new generated query $e_i^q$ and all prior queries $e_j^q$ for $j < i$, using embedding vectors obtained from OpenAI's *text-embedding-ada-002*[1]. The similarity is computed as:

$$\max_{j<i} \text{sim}(\mathbf{e}_i^q, \mathbf{e}_j^q) < \phi, \tag{2}$$

---

[1]https://openai.com/index/new-and-improved-embedding-model/

where $\text{sim}(\cdot, \cdot)$ denotes the cosine similarity function, and $\phi$ is a predefined threshold controlling the allowed similarity. This constraint ensures that the new query is sufficiently different from all previous queries, promoting diversity in exploration.

**Self-Reflection.** If the similarity constraint is not satisfied (i.e., the new query is too similar to previous ones), the `Explorer` engages in self-reflection (Shinn et al., 2024). It discards the current instance and analyzes the reasons for the overlap, adjusting its approach to generate a new query that explores different aspects of the tool. This iterative self-reflection process continues until the `Explorer` produces an instance that meets the diversity criterion.

By incorporating this diversity-promoting exploration strategy, we enhance the exploration coverage of the `Explorer`, allowing it to investigate a broader range of tool functionalities and edge cases. This comprehensive set of exploration instances is crucial for identifying potential misunderstandings and gaps in the tool documentation, thereby providing valuable experiential data for subsequent analysis and documentation rewriting. Figure 3 illustrates our prompt template for `Explorer`.

### 2.3 LEARNING FROM EXPERIENCE

Analogous to how humans learn—acquiring familiarity with new tools through practical experiences and then consulting manuals to deepen understanding—the insights gained during the experience gathering phase provide a foundation for informed and targeted enhancements to the documentation. Thus, building upon the experience gathered in the first phase, the second phase focuses on analyzing this data to refine the tool documentation. In this phase, we introduce an `Analyzer` $\mathcal{M}_A$ designed to identify and address issues within the current tool documentation, thereby guiding the `Rewriter` in making effective revisions.

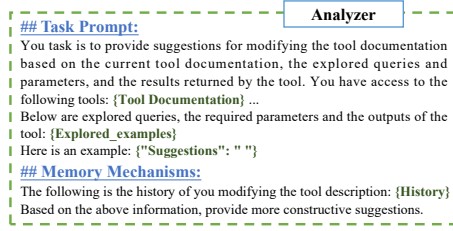

Figure 4: Prompt template for analyzer.

Formally, at the $i$-th iteration, the `Analyzer` $\mathcal{M}_A$ takes the following inputs: current tool documentation $t_{i-1}$, exploration instance $e_i$, tool feedback $r_i$ provided by the `Explorer` $\mathcal{M}_E$, and the history of documentation revisions $\mathcal{T}_i = \{t_j \mid j < i\}$. Then the `Analyzer` $\mathcal{M}_A$ analyzes these inputs to identify issues and generate revision suggestions $s_i$:

$$s_i = \mathcal{M}_A(t_{i-1}, e_i, r_i, \mathcal{T}_i). \tag{3}$$

To ensure that the `Analyzer` can provide high-quality and relevant revision suggestions, we establish several evaluation criteria, including consistency with tool outputs, comprehensiveness, and conciseness without irrelevant information (as detailed in Appendix C). These criteria enable `Analyzer` to identify and assess issues present within the existing tool documentation. By considering the historical evolution of the documentation through the revision history $\mathcal{T}_i$, the `Analyzer` gains valuable insights into past updates, helping it avoid redundant or repetitive suggestions and focus on areas that still require improvement. The prompt template for `Analyzer` is illustrated in Figure 4.

Furthermore, the `Analyzer` delivers its feedback in natural language, offering detailed and nuanced guidance to the `Rewriter` for subsequent updates. This approach contrasts with providing mere scalar feedback, as it ensures the `Rewriter` receives comprehensive insights that facilitate accurate and effective revisions, ultimately enhancing the clarity and usability of the tool documentation.

### 2.4 DOCUMENTATION REWRITING

Building upon the experiences gathered and the revision suggestions obtained from the previous two phases, the final phase focuses on refining the tool documentation to enhance its clarity, accuracy, and usability, ensuring it aligns with the comprehension capabilities of LLMs. This phase also provides suggestions for future exploration directions in the next iteration of the experience gathering phase.

Specifically, we design a `Rewriter` $\mathcal{M}_R$ to synthesize information from the exploration instances $e_i$ and the corresponding tool return results $r_i$ provided by the `Explorer` $\mathcal{M}_E$, as well as the revision suggestions $s_i$ from the `Analyzer` $\mathcal{M}_A$. It is important to note that the `Rewriter` $\mathcal{M}_R$ also takes into account the rewrite history $\mathcal{T}_i$, which includes all previous versions of the tool documentation up to iteration $i$. By integrating these inputs, the `Rewriter` $\mathcal{M}_R$ produces an updated version of the

tool documentation $t_i$ and provides suggestions for the next round of exploration directions $d_i$. This process is formalized as:

$$d_i, t_i = \mathcal{M}_R(t_{i-1}, e_i, r_i, s_i, \mathcal{T}_i). \tag{4}$$

By incorporating the revision history into its process, the `Rewriter` ensures that each version of the documentation builds upon its predecessors, resulting in continuous improvements in clarity, accuracy, and usability. The prompt template to get `Rewriter` is shown in Figure 5.

Furthermore, recognizing that different tools vary in their complexity and the ease with which LLMs can comprehend them (Qin et al., 2023b; Osiurak & Heinke, 2018), we implement a **tool-adaptive termination mechanism** to adaptively determine when to cease modifications for each tool. Analogous to recipes requiring different levels of expertise, some tools may reach optimal documentation faster than others. We consider the iterative process to have converged when there is minimal change between two consecutive versions of the documentation, indicating that the `Rewriter` has sufficiently aligned the documentation with the LLM's understanding.

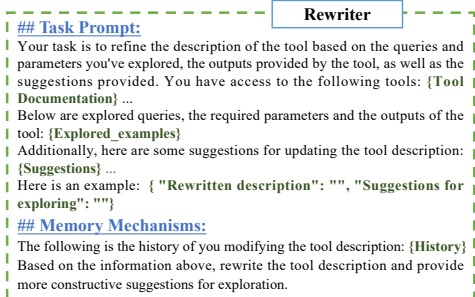

Figure 5: Prompt template for rewriter.

Specifically, inspired by Wieting et al. (2019), we measure the degree of change $\Delta$ between iterations by calculating both the word-match metric (e.g., BLEU score (Papineni et al., 2002)) and the semantic-match metric (e.g., cosine similarity of embeddings):

$$\Delta = \frac{\text{sim}(\mathbf{e}_i^t, \mathbf{e}_{i-1}^t) + \text{BLEU}(t_i, t_{i-1})}{2}, \tag{5}$$

where $\mathbf{e}_i^t$ and $\mathbf{e}_{i-1}^t$ are the embedding vectors of $t_i$ and $t_{i-1}$ obtained using OpenAI's *text-embedding-ada-002*[2]. The function $\text{sim}(\cdot, \cdot)$ calculates the cosine similarity between the semantic embedding vectors of two documentation versions, and $\text{BLEU}(\cdot, \cdot)$ measures the n-gram overlap between them. If $\Delta$ exceeds a predefined termination threshold $\tau$, we stop the iterative modifications.

This tool-adaptive termination mechanism offers several advantages: First, it enhances efficiency by ceasing iterations when the documentation is adequately aligned with the LLM's comprehension, conserving computational resources and time. Second, it prevents unnecessary modifications that could lead to overfitting, thus optimizing the quality of the documentation. By employing both the BLEU score and cosine similarity, we ensure a balanced assessment of structural and semantic alignment, ultimately yielding high-quality documentation tailored for effective LLM utilization.

## 2.5 DRAFT's Strengths

In this section, we outline the key strengths of the proposed DRAFT framework. **First and foremost**, DRAFT operates in a fully automated manner, which significantly reduces resource consumption in comparison to the time-consuming and labor-intensive manual modifications that are typically required. **Furthermore**, by employing a trial-and-error methodology, DRAFT continuously updates tool documentation based on feedback regarding tool usage obtained from LLMs, thereby enhancing the alignment between tool documentation and the operational understanding of LLMs. **Additionally**, DRAFT is capable of dynamically maintaining an accurate and up-to-date representation of evolving features within the tool documentation as the tools develop. It also provides inherent explainability, as the entire process is presented in natural language. Users can easily track the history of modifications and seamlessly integrate expert insights into the updating process.

## 3 Experiments

### 3.1 Experimental Setup

**Datasets.** To verify the effectiveness of DRAFT, we conduct experiments on two benchmarks: ToolBench and RestBench. ToolBench (Qin et al., 2024) is a large-scale benchmark of real-world

---

[2]https://openai.com/index/new-and-improved-embedding-model/

Table 1: Performance comparison of different methods on three datasets. Win% is calculated by comparing each method with ReAct. The term "-" means that EasyTool has not been implemented on the Spotify dataset. The best result for each LLM is in **bold**.

| Model | Method | RestBench-TMDB | | RestBench-Spotify | | ToolBench | |
|---|---|---|---|---|---|---|---|
| | | CP% | Win% | CP% | Win% | CP% | Win% |
| **GPT-4o-mini** | ReAct | 48.00 | 50.00 | 24.56 | 50.00 | 35.00 | 50.00 |
| | DFSDT | 50.00 | 68.00 | 35.08 | 61.40 | 37.00 | 84.00 |
| | EasyTool | 56.00 | 75.00 | - | - | 42.00 | 85.00 |
| | DRAFT (Ours) | **62.00** | **82.00** | **43.85** | **78.94** | **47.00** | **88.00** |
| **Llama-3-70B** | ReAct | 72.00 | 50.00 | 26.31 | 50.00 | 41.00 | 50.00 |
| | DFSDT | 74.00 | 38.00 | 63.15 | 61.40 | 42.00 | 54.00 |
| | EasyTool | 76.00 | **64.00** | - | - | 46.00 | 60.00 |
| | DRAFT (Ours) | **86.00** | **64.00** | **66.66** | **64.91** | **53.00** | **62.00** |
| **GPT-4o** | ReAct | 71.00 | 50.00 | 28.07 | 50.00 | 37.00 | 50.00 |
| | DFSDT | 74.00 | 61.00 | 64.91 | 56.14 | 41.00 | 73.00 |
| | EasyTool | 79.00 | 62.00 | - | - | 45.00 | 77.00 |
| | DRAFT (Ours) | **88.00** | **71.00** | **70.17** | **84.21** | **51.00** | **78.00** |

APIs collected from RapidAPI and BMTools, commonly used to evaluate the capability of LLMs in tool usage. Due to budget constraints, we focus on the most challenging subset of ToolBench, namely I3-Instruction, which contains complex user requests requiring multiple tools from different categories. RestBench (Song et al., 2023) is a benchmark consisting of two real-world scenarios: TMDB, which includes 54 movie-related APIs, and Spotify, which has 40 music-related APIs.

**Evaluation Metrics.** Following previous work (Song et al., 2023; Qin et al., 2024; Yuan et al., 2024), we evaluate performance using two widely adopted metrics: (1) Correct Path Rate (**CP%**), which measures the proportion of instances where the model-generated sequence of tool calls contains the ground truth tool path as a subsequence, allowing for straightforward accuracy assessment. (2) Win Rate (**Win%**), which evaluates effectiveness through pairwise comparisons by a ChatGPT-based evaluator, capturing nuanced performance differences not reflected by rule-based metrics.

**Baselines.** Following Qin et al. (2024) and Yuan et al. (2024), we compare our method with widely adopted baselines, including: (1) ReAct (Yao et al., 2022), which integrates reasoning with action, enabling LLMs not only to justify their actions but also to refine their reasoning processes based on feedback from the environment. (2) DFSDT (Qin et al., 2024), which addresses the issue of error propagation by incorporating a depth-first search strategy to enhance decision-making accuracy. (3) EasyTool (Yuan et al., 2024), which achieves more concise tool descriptions by using ChatGPT to directly rewrite the documentation and incorporate guidelines, thereby enhancing the comprehension of LLMs regarding tool functions and parameter requirements.

**Implementation Details.** For the main experiments, we use the *GPT-4o* as the backbone model for DRAFT, which means that we employ this model to refine the tool documentation by incorporating its own tool usage feedback. We set the similarity threshold $\phi$ to 0.9, termination threshold $\tau$ to 0.75, and maximum iteration count to 5. We select three of the latest LLMs to ensure our evaluation reflects the current state of the field including the closed-source models *GPT-4o* [3] and *GPT-4o-mini* [4], as well as the open-source model *Llama-3-70B* [5]. Our code is available at https://github.com/quchangle1/DRAFT.

## 3.2 EXPERIMENTAL RESULTS

We present our experimental results in Table 1. Based on these results, we have the following observations: While EasyTool can slightly improve experimental performance, it does not incorporate the experience feedback from LLMs to iteratively revise the tool documentation. As a result, it cannot fully align with the understanding of LLMs. In contrast, our method effectively addresses these issues and achieves more significant improvements. We observe that all LLMs achieve better performance

---

[3] https://platform.openai.com/playground/chat?models=gpt-4o-2024-08-06

[4] https://platform.openai.com/playground/chat?models=gpt-4o-mini

[5] https://huggingface.co/meta-llama/Meta-Llama-3-70B

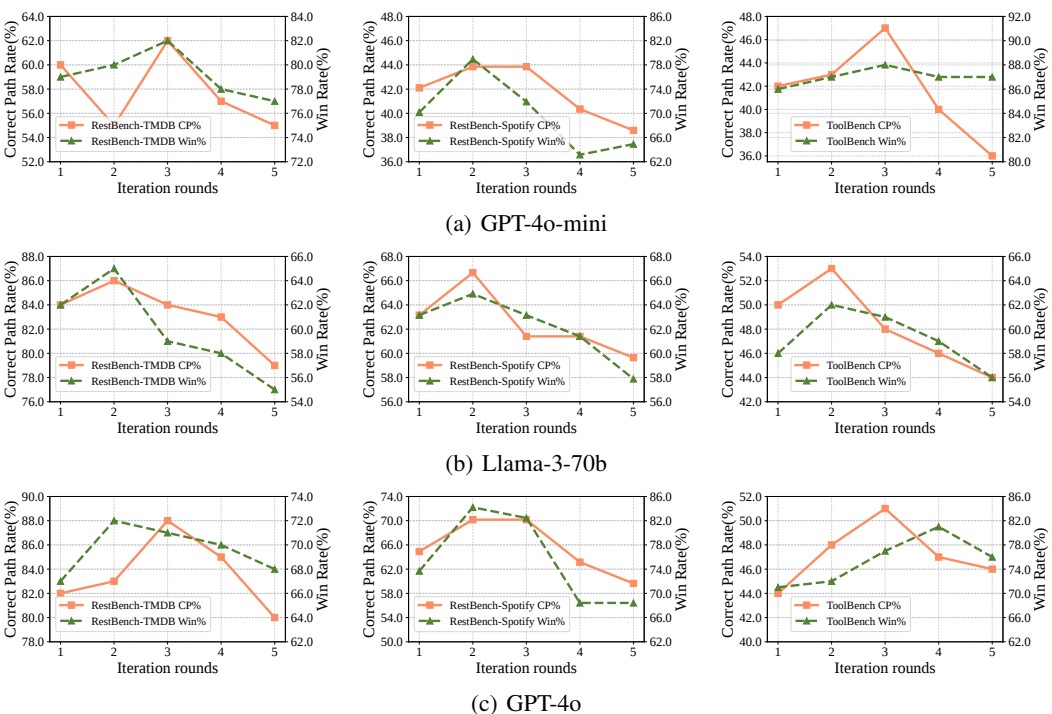

(a) GPT-4o-mini

(b) Llama-3-70b

(c) GPT-4o

Figure 6: A comparative analysis of performance based on varying numbers of iteration rounds.

when using the tool documentation modified by DRAFT. This indicates that although our tool documentation was revised through utilizing tool usage feedback from a single model, it exhibits robust cross-model generalization capabilities. Notably, on the ToolBench dataset, the *GPT-4o-mini* enhanced with DRAFT even surpasses the performance of the *GPT-4o* without DRAFT. These improvements demonstrate the effectiveness of DRAFT, which can be attributed to the fact that DRAFT iteratively learns from the tool usage experiences of LLMs to refine the tool documentation. In this way, DRAFT creates tool documentation specifically aligned with the understanding of LLMs.

## 3.3 FURTHER ANALYSIS

**How does the number of iteration rounds affect the performance of tool learning?** A key feature of DRAFT is its iterative refinement of tool documentation with a tool-adaptive termination mechanism. We examine the effectiveness of these designs by analyzing how the number of iteration rounds influences the performance of downstream tool learning. The results presented in Figure 6 demonstrate a general trend where performance is enhanced with an increasing number of iterations, followed by a subsequent decline. This suggests that iterative modifications are crucial for enhancing tool documentation quality and the ability of LLMs to utilize tools effectively. Such modifications facilitate the exploration of a wider array of examples and the incorporation of additional feedback derived from the tool usage experiences of LLMs, ultimately refining the tool descriptions to achieve superior performance. However, a decline in performance is observed after a certain number of iterations. This decline may be attributed to the introduction of redundant information as the number of iterations increases, potentially leading to overfitting. Therefore, we implement a tool-adaptive termination mechanism to prevent performance degradation and ensure optimal results.

**Does using other models as backbones also ensure cross-model generalization?** Our preliminary experiments have demonstrated that employing *GPT-4o* as the backbone to refine tool documentation by integrating its own usage feedback results in revised documentation that exhibits cross-model generalization, enhancing the

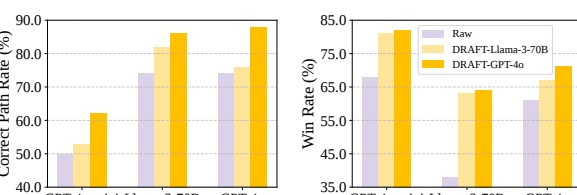

Figure 7: An analysis of cross-model generalization.

performance of other models. This observation prompts

an inquiry into whether using other models as a backbone would similarly yield cross-model generalization. To investigate this, we conduct experiments on RestBench-TMDB using *Llama-3-70B* as a backbone, which led to the generation of a new set of tool documentation informed by its usage feedback. As illustrated in Figure 7, our findings indicate that it also demonstrates cross-model generalization, enhancing the tool usage capabilities across all LLMs. This may result from decoder-only models sharing similar transformer structures and common pre-training corpora, allowing them to achieve consensus on knowledge mastery and needs. Furthermore, we also find that employing *GPT-4o* as a backbone yields superior performance compared to *LLama-3-70B*. This suggests that our method can benefit from continuous improvements in foundational models.

**Are the two mechanisms we proposed truly effective for DRAFT?** We conduct ablation studies on RestBench-TMDB using *GPT-4o* to assess the impact of the two mechanisms incorporated within DRAFT. The results presented in Table 2, highlight the significance of each mechanism: ***w/o diversity*** refers to a variant that the explorer generates exploration instances without applying similarity constraint, meaning that all generated instances are considered. The absence of the diversity-promoting exploration strategy leads to a notable per-

Table 2: Ablation study of the proposed DRAFT.

| Mehtods | TMDB | |
|---|---|---|
| | **CP%** | **Win%** |
| **DRAFT** | **88.00** | **71.00** |
| *w/o* diversity | 84.00 | 69.00 |
| *w/o* adaptive | 80.00 | 68.00 |

formance drop, underscoring the need for its implementation to enhance exploration diversity. ***w/o adaptive*** refers to a variant that does not terminate prematurely based on the iteration conditions; rather, it guarantees that each iteration of the tool documentation completes the designated maximum number of iteration rounds before ceasing operations. The significant decline in performance observed in this variant further emphasizes the benefits of introducing the tool-adaptive termination mechanism, which serves to prevent overfitting while improving efficiency. The results with other LLMs and datasets demonstrate similar trends and are provided in Appendix B.

**Does the modified tool documentation improve the performance of tool retrieval?** In real-world scenarios, there are often numerous tools, making it impractical to input the descriptions of all tools into LLMs (Qu et al., 2024a). Therefore, effective tool retrieval is crucial for downstream tool learning. In this analysis, we evaluate whether the modified tool documentation not only enhances the ability of LLMs to use tools effec-

Table 3: Comparison of tool retrieval performance between raw documentation and our method. We report NDCG@1 and NDCG@10.

| Retriever | Documentation | TMDB | | Spotify | |
|---|---|---|---|---|---|
| | | **@1** | **@10** | **@1** | **@10** |
| BM25 | Raw | 24.0 | 35.0 | **43.9** | 53.9 |
| | DRAFT | **29.0** | **39.4** | **43.9** | **54.2** |
| Contriever | Raw | 29.0 | 40.4 | 45.6 | **49.6** |
| | DRAFT | **31.0** | **44.1** | **47.4** | 49.2 |

tively but also improves the effectiveness of tool retrieval. The results presented in Table 3 indicate that the revised documentation enhances the performance of tool retrieval methods, including the sparse retrieval method BM25 (Robertson et al., 2009) and the dense retrieval method Contriever (Izacard et al., 2021). This demonstrates that the modifications not only improve the readability of the documentation but also enhance its semantic quality, thereby improving the different stages of the whole pipeline for tool learning.

**Does the tool documentation modified by DRAFT also improve human comprehension?** In the previous experiments, we have demonstrated that tool documentation modified by DRAFT can help LLMs better understand and utilize external tools. Next, we aim to verify whether this improvement extends to human understanding of the tools as well. Specifically, we conduct a human evaluation, inviting three well-educated doctor students to evaluate the raw tool documentation and the documentation modified by DRAFT based on three criteria: **completeness**: which documentation is more comprehensive, **conciseness**: which documentation is clearer and more concise, **accuracy**: which documentation reflects the tool's functionality more accurately (details are presented in Appendix A). We randomly sampled 50 cases from RestBench and ToolBench for this evaluation.

The results, as shown in Table 4, indicate that the tool documentation modified by DRAFT demonstrates significant improvements, particularly in terms of completeness and accuracy (some cases are presented in Appendix D). This can be attributed to the fact that DRAFT incorporates feedback from LLMs based on their tool usage experience, which provides execution results during the revision process, ensuring better alignment between the documentation and the actual tool behavior. Moreover,

Table 4: Human evaluation comparing the quality of the raw documentation with ours.

| Dataset | Completeness | | | Conciseness | | | Accuracy | | |
|---------|------|-----|-------|------|-----|-------|------|-----|-------|
| | **Ours** | **Raw** | **Equal** | **Ours** | **Raw** | **Equal** | **Ours** | **Raw** | **Equal** |
| RestBench | 40% | 16% | 44% | 36% | 20% | 44% | 30% | 0% | 70% |
| ToolBench | **Ours** | **Raw** | **Equal** | **Ours** | **Raw** | **Equal** | **Ours** | **Raw** | **Equal** |
| | 68% | 4% | 28% | 56% | 4% | 40% | 56% | 0% | 44% |

our design of a tool-adaptive termination mechanism effectively prevents unnecessary redundancy, leading to more concise documentation. Additionally, the results suggest that ToolBench tools have significantly lower documentation quality than those in the RestBench dataset, as the readability of ToolBench tools exhibited a more substantial improvement after modification by DRAFT. This further highlights the necessity of revising the existing tool documentation.

## 4 RELATED WORK

**Tool Learning.** Recent studies have highlighted the potential of LLMs to utilize external tools in addressing complex problems (Qu et al., 2024b; Wang et al., 2024d). With the aid of external tools, LLMs can obtain up-to-date information (Nakano et al., 2021; Gou et al., 2024a;b), enhance domain-specific knowledge (M. Bran et al., 2024; Zhang et al., 2024a), process multi-modal information (Surís et al., 2023; Gao et al., 2024c), and more. Existing tool learning approaches can be categorized into two types: tuning-based and tuning-free methods (Gao et al., 2024b). Tuning-based methods enhance the tool-using capabilities of LLMs by fine-tuning them on tool-related datasets (Patil et al., 2023; Hao et al., 2024; Yang et al., 2024; Qin et al., 2024; Liu et al., 2024). However, this approach is only applicable to open-source models and requires substantial computational resources. In contrast, tuning-free methods provide LLMs with tool documentation and a few demonstrations (Wei et al., 2022; Hsieh et al., 2023; Paranjape et al., 2023; Du et al., 2024; Shi et al., 2024), relying on the in-context learning ability to understand how to use tools. This approach requires no additional training and allows for the plug-and-play integration of external tools. However, this method necessitates high-quality tool documentation that is aligned with the comprehension of LLMs (Yuan et al., 2024; Chen et al., 2024). In this paper, we propose a method to align with LLMs understanding and improve the quality of tool documentation to enhance the tool-using capabilities of LLMs.

**Learning from Feedback.** Recent studies show that LLMs can improve their initial responses through self-correction, leading to improved performance (Shinn et al., 2024; Madaan et al., 2024; Pan et al., 2024; Huang et al., 2024). However, some researchers observe that relying exclusively on self-correction without external feedback may yield minimal improvements or worsen performance (Zhao et al., 2024). In contrast, incorporating learning from feedback has been shown to improve various tasks (Jin et al., 2023; Gao et al., 2024a; Wang et al., 2024c; Pan et al., 2024; Welleck et al., 2022; Zhang et al., 2024b). The forms of feedback are categorized into scalar and natural language types (Gou et al., 2024a). Scalar feedback provides coarse-grained information and typically serves as a reward signal in reinforcement learning frameworks (Ziegler et al., 2019), while natural language feedback provides detailed insights and is used in prompts for LLMs to enhance performance (Jin et al., 2023). The sources of feedback are diverse, including humans (Ouyang et al., 2022), critic models (Nathani et al.), external tools (Wang et al., 2024b; Qiao et al., 2024), and even the LLM itself. To ensure that tool documentation genuinely reflects the purpose of the tool, we obtain feedback by actually using the tool to get the returned results, thereby producing high quality tool documentation.

## 5 CONCLUSION

In this paper, we highlight that the misalignment between the existing, primarily human-centric tool documentation and the interpretive requirements of LLMs acts as a pivotal barrier obstructing the full potential of tool learning with LLMs. To remedy this, inspired by trial-and-error, we introduce DRAFT, a dynamic and self-improving framework specifically designed to iteratively refine tool documentation based on direct interactions and feedback loops between LLMs and external tools. Through extensive experimentation, our findings substantiate the assertion that our proposed DRAFT markedly enhances the alignment between tool documentation and the operational understanding of LLMs, thereby fostering more effective tool usage.

ACKNOWLEDGEMENTS

This work was funded by the National Key R&D Program of China (2023YFA1008704), the National Natural Science Foundation of China (62472426), PCC@RUC, fund for building world-class universities (disciplines) of Renmin University of China. Work partially done at Engineering Research Center of Next-Generation Intelligent Search and Recommendation, Ministry of Education.

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

APPENDIX

## A    DETAILS OF HUMAN EVALUATION

We recruit three doctoral students familiar with the domain to evaluate the raw tool documentation and its revised versions generated by DRAFT and pay accordingly. They are selected based on their familiarity with the domain and their capability to assess the quality of tool documentation accurately. They are asked to evaluate the documentation through pair-wise comparison based on the following three criteria: 1. Completeness: Which documentation more thoroughly describes the tool's potential uses and key information, such as its usage and parameter descriptions, without missing critical details. 2. Conciseness: Which documentation is clearer and more concise, avoiding irrelevant or redundant information. 3. Accuracy: Which documentation describes the tool functionality more accurately and is free from incorrect information. The Fleiss' Kappa statistics for completeness, conciseness, and accuracy are 0.76, 0.71, and 0.72 on RestBench, while 0.77, 0.69, and 0.71 on ToolBench, indicating a high agreement among the three annotators.

## B    MORE EXPERIMENTS

We also conduct analysis experiments on three datasets using other LLMs. As illustrated in Table 5, both *GPT-4o-mini* and *LLama-3-70B* display trends that are consistent with those observed in *GPT-4o*. The absence of our proposed diversity-promoting exploration strategy and tool-adaptive termination mechanism results in a decline in performance, thereby underscoring the importance of our design innovations. An intuitive explanation posits that, in the absence of the diversity-promoting exploration strategy, the examples generated during each round of exploration may exhibit significant similarity. This similarity could render multiple iterations indistinguishable from a single iteration, thereby undermining the advantages of repeated exploration. Likewise, the absence of the tool-adaptive termination mechanism may result in excessive iterations, which could produce redundant information and contribute to overfitting.

Table 5: An ablation study of the proposed model, DRAFT, was conducted across three LLMs utilizing three distinct datasets.

| Model | Method | RestBench-TMDB | | RestBench-Spotify | | ToolBench | |
|---|---|---|---|---|---|---|---|
| | | CP% | Win% | CP% | Win% | CP% | Win% |
| **GPT-4o-mini** | DRAFT (Ours) | **62.00** | **82.00** | **43.85** | **78.94** | **47.00** | **88.00** |
| | *w/o* diversity | 60.00 | 79.00 | 42.10 | 70.17 | 42.00 | 86.00 |
| | *w/o* adaptive | 55.00 | 77.00 | 38.59 | 57.89 | 36.00 | 87.00 |
| **Llama-3-70B** | DRAFT (Ours) | **86.00** | **64.00** | **66.66** | **64.91** | **53.00** | **62.00** |
| | *w/o* diversity | 84.00 | 62.00 | 63.15 | 63.15 | 48.00 | 61.00 |
| | *w/o* adaptive | 79.00 | 55.00 | 59.64 | 57.89 | 44.00 | 56.00 |
| **GPT-4o** | DRAFT (Ours) | **88.00** | **71.00** | **70.17** | **84.21** | **51.00** | **78.00** |
| | *w/o* diversity | 84.00 | 69.00 | 64.91 | 73.68 | 44.00 | 72.00 |
| | *w/o* adaptive | 80.00 | 68.00 | 59.64 | 68.42 | 46.00 | 76.00 |

## C    DETAILED PROMPTS

Table 6 provides a comprehensive overview of the prompts utilized during the three learning stages of DRAFT, which include experience gathering, learning from experience, and documentation rewriting.

## D    CASE STUDY

### D.1    TOOLBENCH

Table 7 displays the comparison of original tool documentation and modified versions using DRAFT in some cases from the ToolBench dataset. Each case highlights a specific issue found in the raw

documentation, including incompleteness, ambiguity, redundancy, and inaccuracy, all of which are effectively addressed by our proposed method.

## D.2   RESTBENCH

Table 8 displays the comparison of original tool documentation and modified versions using DRAFT in some cases from the RestBench dataset. Although the overall quality of the tools in RestBench is generally superior to that of ToolBench, there remain significant issues, including irrelevance, ambiguity, and incomplete information. This further highlights the necessity of employing our method, which iteratively refines the documentation through interactions with the tools. By systematically addressing these issues, our approach guarantees that the documentation evolves in a way that enhances clarity, relevance, and completeness. This, in turn, minimizes the likelihood of misunderstandings or misuse of the APIs and enhances the ability of LLMs to utilize external tools.

## E   REVISION TRAJECTORY

Table 9 and Table 10 respectively present the revision trajectory of tool documentation through the first, second, and third rounds of modifications using DRAFT. An analysis of these tables reveals that the original tool documentation is characterized by the inclusion of irrelevant information and a notable absence of guidance regarding the necessity of a valid ID, as well as procedures for addressing errors associated with the provision of an invalid ID. These observations underscore significant deficiencies in the original documentation that necessitate revision.

Through an analysis of the revision trajectory, it becomes evident that during the first round, the experience of encountering an invalid ID is documented. The analyzer promptly identifies the issue and provides suggestions for improvement. Subsequently, the rewriter utilizes this feedback to make appropriate updates to the tool documentation, recommending that future exploration should concentrate on retrieving results associated with a valid ID. By the third iteration, the exploration successfully yields results with a valid ID, leading to further updates in the documentation. This process underscores the necessity of conducting multiple iterations. Furthermore, in the absence of a diversity-promoting exploration strategy designed to facilitate diverse exploration, the system may repeatedly encounter similar scenarios, thereby hindering the discovery of valid IDs and obstructing effective updates. This situation emphasizes the critical importance of implementing a diversity-promoting exploration strategy.

Table 11 and Table 12 show the revision trajectories from the fourth and fifth rounds, respectively. Upon analysis, it is evident that as the number of iterations increases, the documentation for the tool becomes increasingly lengthy, leading to the inclusion of redundant information. This redundancy may impede the effective utilization of the tool by LLMs, thereby underscoring the necessity of the tool-adaptive termination mechanism that we have developed.

Table 6: The full prompt for three learning stages of DRAFT.

| **I: Experience Gathering** |
| --- |

*/\* Task prompt \*/*
Your task is to answer the user's query as best you can. You have access to the following tools, which you can use via API call to help with your response: {Tool Documentation}
Now you have the chance to explore the available APIs. You can do this by 1) synthesizing some natural user query that calling the API could help, 2) extracting the parameters needed to call these APIs from the generated query, and 3) Here, you can focus on queries that only require calling the API once.
Now, first input your synthesized user query. You should make the query natural - for example, try to avoid using the provided API descriptions or API names in the query, as the user does not know what APIs you have access to. However, please make sure that the user query you generate includes the parameters required to call the API, for which you need to generate random information. For required parameters like IP address, location, coordinates, etc., provide specific details. For example, instead of simply stating 'an address', provide the exact road and district names. Please note that if the required parameters are ID or username, which you do not know what are valid, you should use the default parameters provided in the API documentation directly. Also try to make the query as specific as possible. Next you need to extract the parameters needed to call the APIs from your generated user queries based on the provided API documentation descriptions.
Here is an example: {"User Query": " ", "Parameters"::{ }}
*/\* Memory Mechanisms \*/*
Below are queries you have already explored: {Explored queries}
Based on these, try to explore queries that can help you understand the API further; Avoid synthesizing queries that are too close to the existing ones. Here are some suggestions to explore the API: {Suggestions}
Now you know a bit more about the API. You can synthesize another user query to explore the API a bit further and consolidate your understanding of the API, based on things that you discovered about this API. You should cherish the opportunity to explore, as each time is precious. Therefore, you should generate new explorations that are different from previous ones as much as possible.

| **II: Learning from Experience** |
| --- |

*/\* Task prompt \*/*
You task is to provide suggestions for modifying the tool documentation based on the current tool documentation, the explored query and parameters, and the results returned by the tool. You have access to the following tools: {Tool Documentation}
Please note that the existing tool documentation may be incomplete or noisy. Previously, you generated some user queries and required parameters to explore this API based on the API documentation. Now, you will be provided with the output of this API under these parameters. You need to consider the following when providing suggestions: For instance, consider whether the current description is consistent with the actual results returned by the tool, whether the description is comprehensive, and whether it is concise and free of irrelevant information. Provide suggestions for modifications based on these aspects. Below are explored queries, the required parameters and the outputs of the tool: {Explored examples}
Here is an example: {"Suggestions": " "}
*/\* Memory Mechanisms \*/*
The following is the history of you modifying the tool description: {History}
Based on the above information, provide more constructive suggestions.

| **III: Documentation Rewriting** |
| --- |

*/\* Task prompt \*/*
Your task is to refine the description of the tool based on the queries and parameters you've explored, the outputs provided by the tool, as well as the suggestions provided. You have access to the following tools: {Tool Documentation}
Please note that the existing tool documentation may be incomplete or noisy. The revised description should focus solely on the functionalities of the API, omitting any irrelevant details. Below are explored queries, the required parameters and the outputs of the tool: {Explored examples}
Based on the feedback provided, here are some guidelines for updating the tool description: {Suggestions}
Due to the limited number of explorations you can perform, you need to value each opportunity. What aspects of this API would you like to explore next? Please provide some suggestions for your next query generation. Just give a direction of the next exploration, don't give a full example of the exploration directly.
Here is an example:{ "Rewritten description": " ", "Suggestions for exploring": " "}
*/\* Memory Mechanisms \*/*
The following is the history of you modifying the tool description: {History}
Based on the above information, provide more constructive suggestions.

Table 7: Comparison of original tool documentation and modified versions using DRAFT in some cases from the ToolBench Dataset.

---

**Raw Tool Description (Incomplete)**
{"Tool Name": "GetSearchImage", "description": " "}

---

**Ours**
{"Tool Name": "GetSearchImage", "description": "Retrieve an image using the specified image identifier and search context ID, with optional parameters for folder and storage."}

---

**Raw Tool Description (Ambiguous)**
{"Tool Name": "il", "description": "Turkish plates. 1 to 81"}

---

**Ours**
{"Tool Name": "il", "description": "Provides location details for Turkish plate numbers based on the plate number parameter, including city, district, neighborhood, and postal code."}

---

**Raw Tool Description (Redundant)**
{"Tool Name": "QR Code Image Generator", "description": "A QR code generator API is a tool that enables developers to generate QR codes within their applications. QR codes are two-dimensional barcodes that can be scanned using a smartphone camera and decoded using QR code reader software. The API allows developers to easily integrate QR code generation functionality into their existing applications, such as mobile apps, web apps, and software.This API can be used to generate QR codes for a variety of purposes, such as:Contact information: Generate a QR code containing your contact information, such as your phone number or email address.Links: Create a QR code that links to a website, YouTube video, or other online content.Payments: Create a QR code for making payments through a mobile wallet or payment app.Events: Generate a QR code for an event, such as a concert or conference, to provide attendees with all the necessary information.Coupons: Create a QR code for a coupon or promo code to be redeemed at a physical store or online.Overall, a QR code generator API is a versatile tool that can help businesses and individuals streamline their processes and improve the user experience for their customers."}

---

**Ours**
{"Tool Name": "QR Code Image Generator", "description": "A QR code generator API is a versatile tool that enables developers to generate QR codes for various purposes, such as contact information, links, payments, events, and coupons. Users should be prepared for potential connectivity issues, including connection timeouts, and are advised to contact the API provider for support in case of errors like 'Connection timed out'. Implementing retry mechanisms and checking network connectivity can help handle such situations effectively."}

---

**Raw Tool Description (Inaccurate)**
{"Tool Name": "SQUAKE", "description": "SQUAKE helps businesses to build sustainable products! With the most performant API in the market, we help travel, mobility, and logistics companies calculate carbon emissions and purchase certified climate contributions in real-time. Implemented within mins!", "Return Results:" "I'm a teapot."}

---

**Ours**
{"Tool Name": "il", "description": "A tool that checks the health status of a specific system and provides relevant information or error messages. It may occasionally return unexpected or humorous responses.", "Return Results:" "I'm a teapot."}

---

Table 8: Comparison of original tool documentation and modified versions using DRAFT in some cases from the RestBench Dataset.

---

**Raw Tool Description (Irrelevant)**
{"Tool Name": "GET_trending_media_type_time_window", "description": "Get the daily or weekly trending items. The daily trending list tracks items over the period of a day while items have a 24 hour half life. The weekly list tracks items over a 7 day period, with a 7 day half life.#### Valid Media Types| **Media Type** | **Description** || - | - || all | Include all movies, TV shows and people in the results as a global trending list. || movie | Show the trending movies in the results. || tv | Show the trending TV shows in the results. || person | Show the trending people in the results. |#### Valid Time Windows| **Time Window** | **Description** || - | - || day | View the trending list for the day. || week | View the trending list for the week. |"}

---

**Ours**
{"Tool Name": "GET_trending_media_type_time_window", "description": "This API provides access to trending items on a daily or weekly basis, including movies, TV shows, or people. The 'media_type' parameter is required and specifies the type of media: 'movie' for movies, 'tv' for TV shows, 'person' for people, or 'all' to include all types. The 'time_window' parameter is also required and can be set to 'day' for daily trends or 'week' for weekly trends. The API response includes a list of trending items with fields such as 'id', 'name', 'original_name', 'media_type', 'popularity', 'gender', 'known_for_department', 'profile_path', and 'known_for'. For each 'known_for' item, details like 'title', 'original_title', 'overview', 'poster_path', 'media_type', 'original_language', 'genre_ids', 'popularity', 'release_date', 'vote_average', and 'vote_count' are included. Pagination is supported, allowing navigation through results using the 'page' parameter, with 'total_pages' and 'total_results' indicating the number of pages and total results available. The API can filter content based on language preferences ('original_language') and content restrictions ('adult')."}

---

**Raw Tool Description (Ambiguous)**
{"Tool Name": "GET_person_person_id_tv_credits", "description": "Get the TV show credits for a person. You can query for some extra details about the credit with the [credit method](#endpoint:xPWdEBLkvCNZSicLN)."}

---

**Ours**
{"Tool Name": "GET_person_person_id_tv_credits", "description": "This API retrieves TV show credits for a person using their unique 'person_id'. It returns two main categories of credits: 'cast' and 'crew'. Each credit includes details such as the TV show's name, first air date, overview, popularity, and the person's role (e.g., Director). The 'crew' category provides additional information like department, job, and episode count. A valid 'person_id' is required, and if an invalid or non-existent 'person_id' is provided, the API will return an error message. The tool does not require any additional parameters. Ensure that the 'person_id' is correctly formatted and exists in the database to avoid errors. Additional details about the TV show credits can be queried using the credit method."}

---

**Raw Tool Description (Error Message)**
{"Tool Name": "GET_tv_latest", "description": "Get the most newly created TV show. This is a live response and will continuously change."}

---

**Ours**
{"Tool Name": "GET_tv_latest", "description": "Retrieve live, comprehensive details about the most recently created TV show, including its name, air dates, episodes, production status, genres, networks, production companies, and countries. The tool also provides information on the latest and upcoming episodes, as well as metadata like popularity and vote statistics. Note that some fields may be empty if the information is not available at the time of the query."}

---

**Raw Tool Description (Incomplete)**
{"Tool Name": "GET_genre_tv_list", "description": "Get the list of official genres for TV shows."}

---

**Ours**
{"Tool Name": "GET_genre_tv_list", "description": "Retrieve the official list of TV show genres from The Movie Database (TMDb). This tool provides a JSON object containing an array of genre objects, each with an 'id' and a 'name'. It is specifically designed for TV show genres and does not require any input parameters."}

---

Table 9: Revision trajectory of tool documentation through the first two iterations of modifications using DRAFT.

| Raw Tool Documentation |
|---|

{"tool name": "GET_person_person_id_tv_credits", "tool description": "Get the TV show credits for a person. You can query for some extra details about the credit with the [credit method](#endpoint:xPWdEBLkvC-NZSicLN)."}

| First Iteration |
|---|

/* Explorer */
{"User Query": "Can you tell me about the TV shows that actor John Doe has been a part of?", "url": "http://api.themoviedb.org/3/person/{person_id}/tv_credits", "Parameters": {}, "API_Response": {"success": false, "status_code": 6, "status_message": "Invalid id: The pre-requisite id is invalid or not found."}}

/* Analyzer */
The tool description should clarify that a valid 'person_id' is required to retrieve TV show credits for a person. The current description does not mention the necessity of this parameter, which is crucial for the tool to function correctly. Additionally, the description should specify that the tool will return an error message if an invalid or non-existent 'person_id' is provided. This will help users understand the importance of providing correct input and what to expect in case of an error.

/* Rewriter */
{ "Rewritten description": "Retrieve TV show credits for a person using their unique 'person_id'. A valid 'person_id' is required to access this information. If an invalid or non-existent 'person_id' is provided, the tool will return an error message indicating the issue. Additional details about the TV show credits can be queried using the credit method.",
"Suggestions for exploring": "To further explore this API, it would be beneficial to understand the structure of the response when a valid 'person_id' is used. This includes the types of information returned about each TV show credit, such as the role played, the show's title, and the dates of involvement. Additionally, exploring how to obtain a valid 'person_id' for a given actor or actress could be useful, as well as understanding any rate limits or restrictions on the number of queries that can be made."}

| Second Iteration |
|---|

/* Explorer */
{"User Query": "What TV shows has actress Jane Smith been involved in throughout her career?", "url": "http://api.themoviedb.org/3/person/{person_id}/tv_credits", "Parameters": {}, "API_Response": {"success": false, "status_code": 6, "status_message": "Invalid id: The pre-requisite id is invalid or not found."}}

/* Analyzer */
The tool retrieves TV show credits for a person using their unique 'person_id'. It is essential to provide a valid 'person_id' to access this information. If an invalid or non-existent 'person_id' is provided, the tool will return an error message with a status code and a status message indicating the issue. The tool does not require any additional parameters. Additional details about the TV show credits can be queried using the credit method. Ensure that the 'person_id' is correctly formatted and exists in the database to avoid errors.

/* Rewriter */
{ "Rewritten description": "This API retrieves TV show credits for a person using their unique 'person_id'. A valid 'person_id' is essential to access the information. If an invalid or non-existent 'person_id' is provided, the API will return an error message with a status code and a status message indicating the issue. The tool does not require any additional parameters. Ensure that the 'person_id' is correctly formatted and exists in the database to avoid errors. Additional details about the TV show credits can be queried using the credit method.",
"Suggestions for exploring": "To further explore this API, consider testing with a known valid 'person_id' to observe the structure and details of a successful response. Additionally, explore how the API handles edge cases, such as a 'person_id' that exists but has no TV credits. Investigate the format and type of additional details that can be retrieved using the credit method to understand the full capabilities of the API."}

Table 10: Revision trajectory of tool documentation in the third round of modifications using DRAFT.

| **Third Iteration** |
| --- |

*/* Explorer */*

{"User Query": "Can you provide the TV show credits for the person with ID 12345?", "url": "http://api.themoviedb.org/3/person/12345/tv_credits", "Parameters": {}, "API_Response": {"cast": [], "crew": [{ "adult": false, "backdrop_path": "/fmPo20ZTzxNketxL2jt6ZX6KSPi.jpg", genre_ids": [10759, 18, 10765,80], "id": 2384, "origin_country": [ "US" ], "original_language": "en", "original_name": "Knight Rider", "overview": "Michael Long, an undercover police officer, is shot while investigating a case and left for dead by his assailants. He is rescued by Wilton Knight, a wealthy, dying millionaire and inventor who arranges life-saving surgery, including a new face and a new identity–that of Michael Knight. Michael is then given a special computerized and indestructible car called the Knight Industries Two Thousand (nicknamed KITT), and a mission: apprehend criminals who are beyond the reach of the law. The series depicts Michael's exploits as he and KITT battle the forces of evil on behalf of the Foundation for Law and Government.", "popularity": 204.362, "poster_path": "/t57NXzTrwp5rnPXiaX4BThrDxt1.jpg", "first_air_date": "1982-09-26", "name": "Knight Rider", "vote_average": 7.5, "vote_count": 1227, "credit_id": "55bb4faec3a3684fc0002cd4", "department": "Directing", "episode_count": 1, "job": "Director"}}

*/* Analyzer */*

The API retrieves TV show credits for a person using their unique 'person_id'. It returns two main categories of credits: 'cast' and 'crew', with details such as the TV show's name, first air date, overview, popularity, and the person's role (e.g., Director). The 'crew' category includes additional information like department, job, and episode count. A valid 'person_id' is required, and if an invalid or non-existent 'person_id' is provided, the API will return an error message. The tool does not require any additional parameters. Ensure that the 'person_id' is correctly formatted and exists in the database to avoid errors. Additional details about the TV show credits can be queried using the credit method.

*/* Rewriter */*

{"Rewritten description": "This API retrieves TV show credits for a person using their unique 'person_id'. It returns two main categories of credits: 'cast' and 'crew'. Each credit includes details such as the TV show's name, first air date, overview, popularity, and the person's role (e.g., Director). The 'crew' category provides additional information like department, job, and episode count. A valid 'person_id' is required, and if an invalid or non-existent 'person_id' is provided, the API will return an error message. The tool does not require any additional parameters. Ensure that the 'person_id' is correctly formatted and exists in the database to avoid errors. Additional details about the TV show credits can be queried using the credit method.", "Suggestions for exploring": "To further explore the API, consider querying for different 'person_id' values to see how the API handles various cases, such as when a person has only cast credits or only crew credits. Additionally, explore how the API responds to edge cases, such as when a person has no TV credits at all. Investigate the structure of the 'cast' category to understand the type of information provided there, as the current exploration focused on the 'crew' category. Finally, test the API's error handling by intentionally using invalid 'person_id' values to observe the error messages returned."}

Table 11: Revision trajectory of tool documentation in the fourth round of modifications using DRAFT.

| Fourth Iteration |
|---|
| */\* Explorer \*/*
{"User Query": "Can you look up the TV show credits for the person with ID 99999 and let me know if they have any crew roles?", "url": "http://api.themoviedb.org/3/person/99999/tv_credits", "Parameters": {}, "API_Response": {"success": false, "status_code": 34, "status_message": "The resource you requested could not be found."}} |
| */\* Analyzer \*/*
The API retrieves TV show credits for a person using their unique 'person_id'. It categorizes credits into 'cast' and 'crew', providing details such as the TV show's name, first air date, overview, popularity, and the person's role. For crew credits, it includes additional information like department, job, and episode count. A valid 'person_id' is required to access this information. If an invalid or non-existent 'person_id' is provided, the API will return an error message with a status code and status message indicating the issue. The tool does not require any additional parameters. Ensure that the 'person_id' is correctly formatted and exists in the database. Additional details about the TV show credits can be queried using the credit method. The API response will include a 'success' flag and, if unsuccessful, a 'status_code' and 'status_message' to indicate the problem. |
| */\* Rewriter \*/*
{"Rewritten description": "This API retrieves TV show credits for a person using their unique 'person_id'. It categorizes credits into 'cast' and 'crew', providing details such as the TV show's name, first air date, overview, popularity, and the person's role. For crew credits, it includes additional information like department, job, and episode count. A valid 'person_id' is required to access this information. If an invalid or non-existent 'person_id' is provided, the API will return an error message with a status code and status message indicating the issue. The tool does not require any additional parameters. Ensure that the 'person_id' is correctly formatted and exists in the database. Additional details about the TV show credits can be queried using the credit method. The API response will include a 'success' flag and, if unsuccessful, a 'status_code' and 'status_message' to indicate the problem.",
"Suggestions for exploring": "To further explore this API, consider querying with a valid 'person_id' to examine the structure and content of a successful response. This will help understand the data fields returned for both 'cast' and 'crew' credits. Additionally, explore edge cases such as querying with a 'person_id' that has only cast credits or only crew credits to see how the API handles these scenarios. Investigating the response for a 'person_id' associated with a highly popular TV show could also provide insights into how popularity metrics are represented."} |

Table 12: Revision trajectory of tool documentation in the fifth round of modifications using DRAFT.

| Fifth Iteration |
|---|

*/\* Explorer \*/*

{"User Query": "Can you look up the TV show credits for the person with ID 24680 and let me know if they have any credits in both cast and crew roles?", "url": "http://api.themoviedb.org/3/person/24680/tv_credits", "Parameters": {}, "API_Response": {"cast": [], "crew": [{ "adult": false, "backdrop_path": "/d1wj4WQ87tj7 dLgXCKfPOe8AOz4.jpg","genre_ids": [18], "id": 19649, "origin_country": ["IT", "GB"], "original_langua- ge": "en", "original_name": "Jesus of Nazareth", "overview": "Dramatizes the Birth, Life, Ministry, Crucifix- ion, and Resurrection of Jesus Christ, largely according to the Holy Bible's New Testament Gospels.", "popu- larity": 38.929, "poster_path": "/dlGmVpZ2rpGRDZsSa64nzyu90xk.jpg", "first_air_date": "1977-03-27", "n- ame": "Jesus of Nazareth", "vote_average": 7.727, "vote_count": 479, "credit_id": "59147af2c3a368423c00e- 100", "department": "Sound", "episode_count": 4, "job": "Sound Recordist"}], "id": 24680}

*/\* Analyzer \*/*

The API retrieves TV show credits for a person using their unique 'person_id'. It categorizes credits into 'cast' and 'crew', providing details such as the TV show's name, first air date, overview, popularity, and the person's role. For crew credits, it includes additional information like department, job, and episode count. A valid 'person_id' is required to access this information. If an invalid or non-existent 'person_id' is provided, the API will return an error message with a status code and status message indicating the issue. The tool does not requi- re any additional parameters. Ensure that the 'person_id' is correctly formatted and exists in the database. Note that the API response does not include a 'success' flag; instead, it provides the 'cast' and 'crew' lists directly. Additional details about the TV show credits can be queried using the credit method.

*/\* Rewriter \*/*

{"Rewritten description": "This API retrieves TV show credits for a person using their unique 'person_id'. It categorizes credits into 'cast' and 'crew', providing details such as the TV show's name, first air date, overvi- ew, popularity, and the person's role. For crew credits, it includes additional information like department, job, and episode count. A valid 'person_id' is required to access this information. If an invalid or non-existent 'p- erson_id' is provided, the API will return an error message with a status code and status message indicating the issue. The tool does not require any additional parameters. Ensure that the 'person_id' is correctly format- ted and exists in the database. The API response does not include a 'success' flag; instead, it provides the 'cast' and 'crew' lists directly. Additional details about the TV show credits can be queried using the credit method.", "Suggestions for exploring": "Consider exploring how the API handles edge cases, such as when a person has no TV credits at all or when they have credits in both cast and crew roles. Additionally, investigate if the API provides any sorting or filtering options for the credits, such as by date or popularity. Another area to explore could be the level of detail provided in the 'overview' field and whether it varies significantly between differ- ent TV shows."}

