# OpenReview forum: "From Exploration to Mastery: Enabling LLMs to Master Tools via Self-Driven Interactions"
_ICLR.cc/2025/Conference — ICLR 2025 Oral_

### Official Review · Reviewer_CrZf · 2024-10-22

**Soundness:** 2
**Presentation:** 3
**Contribution:** 2
**Rating:** 6
**Confidence:** 3

**Summary:**

This work introduces a new prompting workflow for improving tool documentation iteratively called DRAFT. While prior tool documentation is typically iterated on manually and laboriously, this work focuses on automating the process of improving documentation via a three step procedure that can be repeated multiple times. The outputs of prior phases (e.g. Explorer, Analyzer) are fed into subsequent phases (Analyzer, Rewriter), where each phase produces prior experiences and filtering that is then used to refine the prompt further. To encourage diverse exploration and ensure that DRAFT terminates at some point, the authors introduce new strategies that encourage such convergence. Following the experimental setup of prior works, the authors show that DRAFT outperforms previous interactive prompting methods and run ablations demonstrating the effect of different parameters and effects of DRAFT.

**Strengths:**

The motivation established by the authors is sound and makes sense. In addition, the formalizations introduced in Section 2 make it easy to follow how the outputs on one prompt feed into the next. The use of an exploration phase is interesting, similar to fuzzing of a program for software engineering tests. The experiments are sound and it follows prior works. The related work covered by the authors is quite comprehensive. The ablations are complete, as the authors do a good job examining the several parameters of DRAFT.

**Weaknesses:**

- While they are clearly thoughtfully made, I would recommend Figures 1 and 2 be simpler to make it easier for a reader to understand what is going on. The captions would also benefit from being more specific about what is going on in the diagram. For instance, for Figure 2, I did not understand what “Trail” (I think this was supposed to be “Trial”?) refers to, is it an iteration of explorer/analyzer/rewriter?
- The methodology section, while clear in how the symbols relate to one another, feels somewhat ungrounded in terms of why certain equations or prompt templates were used. For instance, on line 299 - “when the documentation is adequately aligned with comprehension” - what does this mean? What is adequately aligned? I think introducing more citations and explanations for why contributions such as the diversity and termination mechanisms would be needed for a more rigorous justification of why this method is interesting, and not just a prompt based workflow that happens to work.
- DSPy (Khattab 2022) performs prompt optimization with multiple rounds of trial/error against a validation set. What makes DRAFT different from DSPy as a prompt optimization technique?
- Although the method is proven to be effective in the experimental results, I think it would be nice to see, qualitatively, the effects of better documentation on downstream performance. While it seems that DRAFT improves the quality of the documentation, it is not obvious why better documentation improves empirical performance. For instance, it may also be possible that more “concise” documentation does not necessarily lead to improvements in downstream performance.

**Questions:**

- Equation 1: What is M? Is this supposed to be A, the Explorer?
- Line 209: How do the two components come together to make up the diversity promoting exploration strategy? Two different constraints were explained, but how do they come together to promote diversity? Is this explicitly included in Algorithm 1 somewhere? Also, is there any citation or support for why self-reflection would promote diversity? How do we know this works?
- Equation 3: Again, what is M?
- Line 281 - “Analogous to recipes requiring different levels of expertise, some tools may reach optimal documentation faster than others”. Any citation for this? What is “optimal” in this case? Also, what does faster mean? Is it in terms of the number of trials? Or the number of explorations needed?
- Equation 5: How did you come up with this? Ignoring empirical performance, why does similarity score + BLEU score work for measuring the degree of change? Why are they weighted equally (e.g. why not weigh surface form similarity / BLEU higher than semantic similarity?)
- Experimental Setup: How many rounds of DRAFT iterations were run (did DRAFT ever not terminate within a certain number of steps)? Were the initial documentations the original ones from DFSDT or ReAct?
- Typo Line 326 - “EayTool”
- The human study detailed in Line 473 onwards feels like it could be better explained. Are the doctor-al (Line 477) students part of the author group? What examples or guidelines did the human annotators use to judge the three criteria correctly? Did the human annotators generally agree with each other (Fleiss Kappa score)?
- Section 2 read a bit repetitive at times - was this section written with the assistance of a language model? It may be easier to read with less redundancy. Specifically, the beginnings of each of the subsections essentially repeat what was stated in the introduction and prior subsections.

---

> ### Author Response · Authors · 2024-11-19
> **Response to Reviewer CrZf : Part 1**
>
> We sincerely thank the reviewer for the constructive comments and suggestions, which are very helpful for improving our paper. The revisions have been incorporated in the revised manuscript marked in blue. Please kindly find point-to-point responses below.
> > [W1] While they are clearly thoughtfully made, I would recommend Figures 1 and 2 be simpler to make it easier for a reader to understand what is going on. The captions would also benefit from being more specific about what is going on in the diagram. For instance, for Figure 2, I did not understand what "Trail" (I think this was supposed to be "Trial" ?) refers to, is it an iteration of explorer/analyzer/rewriter?
>
> **Response:** We thank the reviewer for the insightful suggestion. We apologize for any confusion caused and have now added more specific explanations in the captions of each figure to help readers better understand the content of each part. Regarding the term "Trail" in Figure 2, we appreciate your attention to detail-it was a typographical error, and we intended to use "Trial", referring to an iteration of the explorer/analyzer/rewriter process. Because our method follows a trial-and-error approach, we denote each iteration in the figure as a "trial". We have **corrected this in the revised version** and provide further clarification to avoid confusion. Thanks again for pointing this out.
>
> > [W2] The methodology section, while clear in how the symbols relate to one another, feels somewhat ungrounded in terms of why certain equations or prompt templates were used. For instance, on line 299 - "when the documentation is adequately aligned with comprehension" - what does this mean? What is adequately aligned? I think introducing more citations and explanations for why contributions such as the diversity and termination mechanisms would be needed for a more rigorous justification of why this method is interesting, and not just a prompt based workflow that happens to work.
>
> **Response:** We thank the reviewer for the valuable question and apologize for any confusion caused. Regarding the question, "what does this mean? What is adequately aligned?", our approach involves iteratively updating the tool documentation based on feedback from the LLM's interactions with the tool and the tool's output. However, not all feedback collected during these iterations is beneficial for improving the tool documentation. For example, once **the documentation has incorporated sufficient feedback and has been refined to a high quality,** additional feedback from further LLM interactions may no longer provide additional value. At this point, we consider the documentation to be "adequately aligned with comprehension." In our implementation, we evaluate the similarity between documentation versions across iterations. When **the similarity between two consecutive versions exceeds a specified threshold**, we assume that further updates will yield minimal improvements.
>
> Regarding the concern "I think introducing more citations and explanations..." We thank the reviewer for the valuable suggestion. We have **introduced additional citations on line 206 in the revised manuscript. Diversity in exploration has been shown to be both effective and necessary across multiple fields.** For example, Hong et al. [1] demonstrate that in deep reinforcement learning, when the state space is large, ensuring diversity in exploration significantly improves learning outcomes. Additionally, Friedrich et al. [2] show that diversity is essential for global exploration in evolutionary algorithms. In our method, a diversity-promoting exploration strategy is crucial, as it enables us to capture a **broader range of usage scenarios and feedback types** to refine the documentation, thus avoiding overfitting to a single type of feedback. The termination mechanism, on the other hand, is necessary to prevent unnecessary iterations when the documentation has reached a sufficient quality level. It not only **optimizes efficiency** but also aligns with findings in iterative refinement processes, where stopping criteria are crucial to **avoid diminishing returns in performance improvements** [3-5]. Additionally, our **ablation study in Section 3.3** further demonstrates the effectiveness of these two proposed mechanisms.
> We will clarify this in the manuscript to help make our approach more transparent. Thank you for highlighting this point.
>
> [1] Hong Z et al. Diversity-driven exploration strategy for deep reinforcement learning, Neurips 2018
>
> [2] Friedrich T et al. Analysis of diversity-preserving mechanisms for global exploration
>
> [3] Madaan A et al. Self-Refine: Iterative Refinement with Self-Feedback, Neurips 2023
>
> [4] Tandon N et al. Self-Correction: Repairing Model Output Errors After Deployment Using a Dynamic Memory of Feedback, NAACL 2022
>
> [5] Stiennon N et al. Learning to summarize from human feedback, Neurips 2020

---

> ### Author Response · Authors · 2024-11-19
> **Response to Reviewer CrZf : Part 2**
>
> > [W3] DSPy (Khattab 2022) performs prompt optimization with multiple rounds of trial/error against a validation set. What makes DRAFT different from DSPy as a prompt optimization technique?
>
> **Response:** We thank the reviewer for raising this question. We would like to clarify the differences between DSPy and DRAFT. While both DSPy and DRAFT utilize a trial-and-error approach, their **goals and tasks are quite different.** DSPy is designed to construct and optimize complex LM pipelines using modular and declarative programming, focusing especially on transferability across tasks and enhancing trial-and-error efficiency through compilation optimization [1]. In contrast, DRAFT is designed to dynamically adjust and optimize tool documentation based on the interaction feedback between LLMs and external tools, which significantly bridges the gap between them by enabling the LLMs to better comprehend and utilize the tools at their disposal, thereby enhancing the overall tool-using capabilities of LLMs.
> Furthermore, the **objects and methods of optimization differ**. DSPy optimizes prompt generation modules through a **training-based** approach, whereas DRAFT centers on **tuning-free optimization of tool documentation.** Our method also introduces a diversity-promoting exploration strategy and a tool-adaptive termination mechanism to enhance this process. Additionally, prompt optimization techniques like those in DSPy **could be incorporated into DRAFT** to help create more effective prompts and further improve performance.
>
> [1] Khattab O et.al DSPy: Compiling Declarative Language Model Calls into Self-Improving Pipelines, ICLR 2024
>
> > [W4] Although the method is proven to be effective in the experimental results, I think it would be nice to see, qualitatively, the effects of better documentation on downstream performance. While it seems that DRAFT improves the quality of the documentation, it is not obvious why better documentation improves empirical performance. For instance, it may also be possible that more "concise" documentation does not necessarily lead to improvements in downstream performance.
>
> **Response:** We thank the reviewer for raising this concern. Intuitively, the LLM relies on tool documentation to understand tool information and usage [1]. If the documentation contains **noise or irrelevant information, it may mislead the LLM, resulting in incorrect tool usage** and decreased performance. Well-designed documentation, which includes more relevant information, helps the LLM better understand the tool and its usage, **increasing the likelihood of selecting the correct tool and filling in necessary parameters accurately**, thus enhancing performance.
>
> When designing our model, we considered that the LLM's perception of better documentation might differ from human expectations. To bridge the gap between the existing tool documentation and the LLM's needs, we refine the documentation based on **feedback from the LLM's interactions** with the tool, resulting in documentation aligned with the LLM's requirements. We agree that conciseness alone may not always lead to improved performance; however, **DRAFT's iterative approach focuses not only on conciseness but also on relevance and clarity**, enabling the documentation to better align with the LLM's comprehension.
>
> Additionally, the experiments in **Section 3.2** comparing the LLM's use of **original versus DRAFT-refined documentation** further demonstrate that the improved documentation enhances the LLM's tool usage capability. We present the experimental results below in a clearer manner for better understanding:
>
> | Model | Documentation | RestBench-TMDB |  | RestBench-Spotify |  | ToolBench |  |
> | :---: | :---: | :---: | :---: | :---: | :---: | :---: | :---: |
> |  |  | CP% | Win% | CP% | Win% | CP% | Win% |
> | GPT-4o-mini | Raw | 50.00 | 68.00 | 35.08 | 61.40 | 37.00 | 84.00 |
> |  | DRAFT | 62.00 | 82.00 | 43.85 | 78.94 | 47.00 | 88.00 |
> | Llama-3-70B | Raw | 74.00 | 38.00 | 63.15 | 61.40 | 42.00 | 54.00 |
> |  | DRAFT | 86.00 | 64.00 | 66.66 | 64.91 | 53.00 | 62.00 |
> | GPT-4o | Raw | 74.00 | 61.00 | 64.91 | 56.14 | 41.00 | 73.00 |
> |  | DRAFT | 88.00 | 71.00 | 70.17 | 84.21 | 51.00 | 78.00 |
>
> We can find that across all models and datasets, tool documentation modified by DRAFT consistently outperforms raw tool documentation in downstream tasks.
>
> [1] Hsieh C et.al Tool Documentation Enables Zero-Shot Tool-Usage with Large Language Models
>
> [2] Yuan S et.al EASYTOOL: Enhancing LLM-based Agents with Concise Tool Instruction

---

> ### Author Response · Authors · 2024-11-19
> **Response to Reviewer CrZf : Part 3**
>
> > [Q1] Equation 1: What is M? Is this supposed to be A, the Explorer?
>
> **Response:** We apologize for the confusing of the typo. On line 189, we mistakenly referred to $ \mathcal{M}_E $ as $ \mathcal{A}_E $.  To clarify,  $ \mathcal{M}_E $ denotes the _Explorer_,  $ \mathcal{M}_A $ represents the _Analyzer_, and  $ \mathcal{M}_R $ signifies the _Rewriter_. We ensure this is **corrected on line 189** in the revised manuscript.  Thanks again for point out this issue.
>
> > [Q2] Line 209: How do the two components come together to make up the diversity promoting exploration strategy? Two different constraints were explained, but how do they come together to promote diversity? Is this explicitly included in Algorithm 1 somewhere? Also, is there any citation or support for why self-reflection would promote diversity? How do we know this works?
>
> **Response:** We thank the reviewer for the thoughtful and detailed questions. Below are our point-by-point clarification:
>
> When simulating new tool usage scenarios and generating queries, we first evaluate whether the newly generated query satisfies the **Similarity Constraint**. If the constraint is met, there is no need for further self-reflection. However, as stated on Line 219, when the query does not meet the Similarity Constraint, we provide both the prior queries and newly generated query to the LLM, prompting it to engage in **Self-Reflection**. This process helps the LLM analyze why the constraint was not satisfied and refine its query generation accordingly. In this way, the **Similarity Constraint acts as a conditional control, while** **Self-Reflection functions as a method**. Together, these two components interact to promote diversity in the exploration process.
>
> Regarding the question "Is this explicitly included in Algorithm 1?": Yes, these components are described in **Lines 6 and 7 of Algorithm 1.** Line 6 outlines the Similarity Constraint as a conditional control mechanism, while Line 7 describes the Self-Reflection process.
>
> For the question "Is there any citation or support for why self-reflection would promote diversity? How do we know this works?": Self-reflection has demonstrated its effectiveness across multiple domains. For example, Shinn et al.[1] showed that by enabling LLMs to learn from erroneous information through self-reflection, their performance **improved in decision-making, reasoning, and programming tasks.** In our case, when the generated query fails to satisfy the Similarity Constraint, it essentially **represents erroneous or suboptimal information**. Providing this feedback to the LLM through Self-Reflection helps it **produce queries that meet the constraint while fostering diversity.** We have included these citations on line 220 in the revised manuscript to reinforce the foundation for our approach. Additionally, the **ablation studies in Section 3.3** provide empirical evidence for the effectiveness of the diversity-promoting exploration strategy. The results show that this strategy significantly contributes to improving the downstream performance.
>
> [1] Shinn N et al. Reflexion: Language agents with verbal reinforcement learning, Neurips 2023.
>
> > [Q3] Equation 3: Again, what is M?
>
> **Response:** We thank the reviewer for pointing out the typo again. We apologize for any confusion caused. As clarified earlier, $ \mathcal{M}_E $ denotes the _Explorer_,  $ \mathcal{M}_A $ represents the _Analyzer_, and  $ \mathcal{M}_R $ signifies the _Rewriter_. We ensure this is **corrected** in the revised manuscript.

---

> ### Author Response · Authors · 2024-11-19
> **Response to Reviewer CrZf : Part 4**
>
> > [Q4] Line 281 - "Analogous to recipes requiring different levels of expertise, some tools may reach optimal documentation faster than others". Any citation for this? What is "optimal" in this case? Also, what does faster mean? Is it in terms of the number of trials? Or the number of explorations needed?
>
> **Response:** We thank the reviewer for raising these thoughtful questions. Here are our detailed explanations:
> 1. For the question "Any citation for this?": Intuitively, as in the real world, the effort required for humans to master different tools varies. For example, learning to ride a bicycle typically requires less effort than learning to drive a car. Similarly, **Qin et al. [1] and Osiurak et al. [2] categorized tool use into three levels of intelligence**, showing that the difficulty of using tools increases progressively across these categories. We have included citations on line 278 to support our claim.
> 2. For the question "What is 'optimal' in this case?": Our approach involves iteratively refining the tool documentation. In this context, "optimal" refers to the point at which further iterations **using the feedback no longer significantly improve the documentation.** Specifically, we measure the similarity between documentation versions across iterations, and when the **similarity between two consecutive versions exceeds a predefined threshold**, we consider the documentation to have **reached its optimal state**, as further updates are unlikely to yield meaningful improvements.
> 3. For the question "What does faster mean?": "Faster" refers to the **number of iterations required to reach the optimal state**, which corresponds to the number of trials or explorations conducted. Each iteration includes an exploration phase, and simpler tools often require fewer iterations to refine their documentation effectively. For instance, a straightforward tool may reach its optimal state after two iterations, while a more complex tool with intricate parameters or edge cases may require several iterations. Thus, **simpler tools are considered to reach the optimal state faster than more complex ones.**
>
> [1] Qin Y et.al Tool Learning with Foundation Models, ACM Computing Surveys.
>
> [2] Osiurak F et.al Looking for intoolligence: A unified framework for the cognitive study  of human tool use and technology, American Psychologist.
>
> > [Q5] Equation 5: How did you come up with this? Ignoring empirical performance, why does similarity score + BLEU score work for measuring the degree of change? Why are they weighted equally (e.g. why not weigh surface form similarity / BLEU higher than semantic similarity?)
>
> **Response:** We thank the reviewer for raising these insightful questions. Here are our detailed clarification:
> 1. For the question "How did you come up with this?": Equation 5 was designed to quantify changes in tool documentation between iterations by balancing semantic similarity and surface-level textual similarity. This combination ensures that **both meaning and structure** are reflected in the metric. Using only BLEU or cosine similarity independently could lead to inaccuracies. For example, the sentences "I enjoy reading books on artificial intelligence and machine learning." and "I like studying materials about AI and ML." have a cosine similarity of 0.9383 but a BLEU score of 0.0264, showing significant divergence. Similarly, BLEU can be high while cosine similarity is low. By combining both measures, we mitigate the limitations of each, capturing a more accurate representation of changes in tool documentation.
> 2. For the question "Ignoring empirical performance, why does similarity score + BLEU score work for measuring the degree of change?": This combination **has been shown to be effective** in other contexts. For instance, Wieting et al.[1] adopted a similar approach, combining semantic similarity measures (e.g., SIMILE) with BLEU score. Their results demonstrated that this combination effectively achieves better performance compared to using BLEU alone. In our case, this dual-focus approach **ensures a balanced assessment of both structural and semantic alignment in tool documentation.** We have added this citation on line 289.
> 3. For the question "Why are they weighted equally?":  In the **absence of strong empirical evidence favoring one component over the other**, we chose to weight the two scores equally as a starting point. This provides a simple yet reasonable way to balance semantic integrity and textual changes. Additionally, we have **achieved better downstream experimental results under this equal weighting**, and the **budget constraints further limited additional fine-tuning of these weights**. However, we recognize that in specific scenarios, one factor might be more critical than the other. This flexibility could be **explored in future studies.**
>
> [1] Wieting J et.al Beyond BLEU: Training Neural Machine Translation with Semantic Similarity, ACL 2019.

---

> ### Author Response · Authors · 2024-11-19
> **Response to Reviewer CrZf : Part 5**
>
> > [Q6] Experimental Setup: How many rounds of DRAFT iterations were run (did DRAFT ever not terminate within a certain number of steps)? Were the initial documentations the original ones from DFSDT or ReAct?
>
> **Response:** We thank the reviewer for the detailed question. For the question "How many rounds of DRAFT iterations were run (did DRAFT ever not terminate within a certain number of steps)?": As shown in our analysis in Section 3.3 on performance variation across iteration rounds, we found that performance generally declined after the fifth round across nearly all datasets and models. Therefore, we set the **maximum iteration count to 5**. Additionally, if the degree of change in tool documentation between iterations exceeds a predefined termination threshold, the process will **terminate early.** Thus, the stopping conditions are **either reaching the maximum of 5 iterations or satisfying the predefined early stopping condition**. We have included this on line 365 in the revised manuscript.
>
> For the question "Were the initial documentations the original ones from DFSDT or ReAct?": We apologize for any confusion. In fact, **ReAct and DFSDT are both tool learning methods that do not modify the original documentation.** EasyTool, on the other hand, is a method that involves modifying the documentation. Our approach **begins with the original, unaltered documentation** to ensure a fair comparison. Following Yuan et al.[1], both DRAFT and EasyTool utilize DFSDT as the tool learning method, with the difference being that each uses its respective modified version of the documentation. ReAct and DFSDT both utilize the original tool documentation.
>
>
> [1] Yuan S et.al Easytool: Enhancing llm-based agents with concise tool instruction
>
> > [Q7] Typo Line 326 - "EayTool"
>
> **Response:** We thank the reviewer for pointing out this typo. In line 326, "EayTool" should indeed be corrected to "EasyTool." We ensure this is updated in the revised manuscript.
>
> > [Q8] The human study detailed in Line 473 onwards feels like it could be better explained. Are the doctor-al (Line 477) students part of the author group? What examples or guidelines did the human annotators use to judge the three criteria correctly? Did the human annotators generally agree with each other (Fleiss Kappa score)?
>
> **Response:** We thank the reviewer for this valuable suggestion. In the revised manuscript, we have included more detailed explanations on **Appendix A** to address these concerns:
>
> 1. For the question "Are the doctor-al (Line 477) students part of the author group?": The annotators involved in this evaluation **were not part of the author group**. Specifically, we recruited three doctoral students familiar with the domain to evaluate the raw tool documentation and its revised versions generated by DRAFT and paid accordingly. They **were selected based on their familiarity with the domain and their capability to assess the quality of tool documentation accurately.**
>
> 2. For the question "What examples or guidelines did the human annotators use to judge the three criteria correctly?": We provided the annotators with the documentation of the tool and a sample of the tool's output. As stated on Line 478, they were then asked to evaluate the documentation based on the following three criteria: 1. **Completeness**: Whether the tool documentation thoroughly describes the tool's potential uses and key information, such as its usage and parameter descriptions, without missing critical details. 2. **Conciseness**: Whether the documentation is clear and concise, avoiding irrelevant or redundant information. 3. **Accuracy**: Whether the documentation's described tool functionality aligns with its actual output and is free from incorrect information.
>
> 3. For the question "Did the human annotators generally agree with each other (Fleiss Kappa score)?": We computed the Fleiss Kappa score to measure the agreement among the three annotators. The results are as follows:
>
> | Dataset | Completeness | Conciseness | Accuracy |
> | :---: | :---: | :---: | :---: |
> | RestBench | 0.76 | 0.71 | 0.72 |
> | ToolBench | 0.77 | 0.69 | 0.71 |
>
> These results indicate a **high level of agreement across all three criteria on both datasets**, demonstrating the reliability of the evaluations.
>
> We hope this clarification addresses the reviewer's concerns. Further details and these results have been incorporated into the revised manuscript.

---

> ### Author Response · Authors · 2024-11-19
> **Response to Reviewer CrZf : Part 6**
>
> > [Q9] Section 2 read a bit repetitive at times - was this section written with the assistance of a language model? It may be easier to read with less redundancy. Specifically, the beginnings of each of the subsections essentially repeat what was stated in the introduction and prior subsections.
>
> **Response:** We thank the reviewer for this helpful feedback. We apologize if the repeated emphasis on certain points affected the readability of Section 2. Some of the content was indeed mentioned briefly in the introduction, but it was **expanded** in Section 2 to clarify each module's **motivation** and ensure **continuity** throughout the paper. We believe that, while there is some repetition, these reiterations of key information aid readers in fully understanding our design motivations and contribute to a more coherent flow. All content in this section was originally written by the authors; however, we used ChatGPT to assist with grammar refinement for clearer presentation, as stated during submission in accordance with the submission guidelines. Thank you for highlighting this, and we will review Section 2 to ensure the repetition is both purposeful and concise.
>
> **We hope the above response will fully address your concerns about our work.** We really appreciate your insightful and constructive comments to further help us improve the quality of our manuscript. Thanks again! Please do not hesitate to let us know if you have any further questions.

---

> ### Author Response · Authors · 2024-11-27
> **A kind reminder: revision deadline is approaching**
>
> Dear Reviewer CrZf,
>
> First of all, we would like to sincerely thank you for taking the time to review our paper and provide valuable feedback. Your comments have been incredibly helpful in improving the content of the paper.
>
> As we are currently in the rebuttal stage, we would kindly like to remind you that if you have any further suggestions or feedback on our response, we would greatly appreciate it if you could share them by **the 27th**. After this date, we will no longer be able to make modifications to the PDF based on the reviewers' comments. Your continued guidance is crucial for us to refine the paper.
>
> Once again, thank you for your hard work and support, and we look forward to your valuable response.
>
> Best regards,
>
> Authors

---

> ### Author Response · Authors · 2024-12-02
> **Deadline is approching**
>
> **Dear Reviewer CrZf**,
>
> As the discussion deadline will end in less than two days, we would like to know whether our responses have adequately addressed your concerns. Please do not hesitate to reach out if you have any further questions or need clarification before the deadline. If you find our response satisfactory, we would be grateful if you could consider revising the score.
>
> Thank you once again for your insightful guidance.
>
> Best regards,
>
> Authors

---

> > ### Comment · Reviewer_CrZf · 2024-12-02
> > **Response to Authors**
> >
> > Thanks to the authors for their very thorough responses, I greatly appreciate all the corrections and answers to my questions. I've decided to raise my score from 3 to 6. I think much of my feedback has been addressed. My remaining feedback for this paper is just that the details for how to make DRAFT work for other domains / benchmarks is clear, it seems very cool and I look forward to trying it out for other agentic tasks beyond Toolbench and Restbench.

---

> > > ### Author Response · Authors · 2024-12-03
> > > **Thanks for your thoughtful feedback and for raising your score!**
> > >
> > > Thanks for your positive feedback and for raising your score! We sincerely appreciate your recognition of our efforts to address the major concerns. We will explore the application of DRAFT to other agentic tasks in our future work. Thank you again for your valuable feedback and support!
> > >
> > > Best regards,
> > >
> > > Authors

---

### Official Review · Reviewer_JQEK · 2024-11-01

**Soundness:** 4
**Presentation:** 3
**Contribution:** 4
**Rating:** 8
**Confidence:** 4

**Summary:**

This paper addresses the tool usage challenge for LLMs. The core idea is to develop a explore-analyze-rewrite approach to refine tool documentation to improve LLMs' ability to use tools that may have lower quality documentation. To enhance tool documentation refinement, DRAFT encourages diverse queries to try out different queries to trigger different tool behaviors; these execution results are then leveraged into the analysis phase for documentation rewriting.

DRAFT is evaluated on ToolBench and RestBench, results on CP% and WIN% shows that the refined document improves task completion, especially compared to closely related EasyTool. Detailed analysis of iteraction rounds, ablation results on diversity and adaptive shows significance of features in DRAFT. Also, it's interesting to learn that refined document could also potentially benefit tool retrieval and human understanding.

**Strengths:**

This paper provides a well designed approach for improving LLMs' tool using ability without additional training. The document refinement approach, while not new, is well designed considering how to leverage LLMs' self-reflection ability and trial-and-error approach to enhance document quality. This paper has the following strengths:

1. Leveraging diversity of exploration and trial-and-error to compensate potential limitation of the the initial document. Where other refinement approach like EasyTool may not be able to obtain such extra information.
2. The evaluation has fair comparison with related work to highlight performance gain. The detailed analysis show additional benefits of this approach.

**Weaknesses:**

This paper can be improved upon the following two areas:

1. Please elaborate more in detail about Explorer design and evaluation. DRAFT relies on diverse queries/parameters to trigger different tool behaviors. It looks like the query generation depends on LLM to sample parameters automatically based on tool signature as provided by toolbench. It would be ideal to analyze how diverse in terms of parameter coverage, especially for certain types of parameter like arrays / strings. I imagine if the document is indeed incomplete as mentioned in the intro, these special cases may be less likely going to be hit during exploration. While existing benchmarks conveniently provide well defined parameter domain, it would worth the authors to include in the discussion about how to generalize this for tool documents without good parameter documentation.

2. The exploration-revision loop essentially helps the system to collect more input-output examples that previously doesn't exist in the document. A qualitative analysis between examples generated by revision-only based tool like EasyTool and new examples generated by DRAFT would be helpful to understand this nuance difference.

3. The intro motivates different types of documentation limiations (incomplete, redundant, inaccurate), it would be good to provide qualitative feedback about which scenarios DRAFT is best at improving, especially comparing to prior tools.

**Questions:**

Check above. In general, this paper is well written, and the experiments are quite convincing to me. Please check above for clarifications about (1) parameter space exploration, (2) analysis of example quality and (3) types of documentations DRAFT best at targeting comparing to others.

---

> ### Author Response · Authors · 2024-11-19
> **Response to Reviewer JQEK : Part 1**
>
> We sincerely thank the reviewer for the constructive comments and suggestions, which are very helpful for improving our paper. We are also grateful that you recognized the strengths of our paper. The revisions have been incorporated in the revised manuscript marked in blue. Please kindly find point-to-point responses below.
>
> > [W1] Please elaborate more in detail about Explorer design and evaluation. DRAFT relies on diverse queries/parameters to trigger different tool behaviors. It looks like the query generation depends on LLM to sample parameters automatically based on tool signature as provided by toolbench. It would be ideal to analyze how diverse in terms of parameter coverage, especially for certain types of parameter like arrays / strings. I imagine if the document is indeed incomplete as mentioned in the intro, these special cases may be less likely going to be hit during exploration. While existing benchmarks conveniently provide well defined parameter domain, it would worth the authors to include in the discussion about how to generalize this for tool documents without good parameter documentation.
>
> **Response:** We thank the reviewer for this valuable comment and hope the following clarification addresses your concerns.
>
> Regarding the question "Please elaborate more in detail about Explorer design and evaluation.": the Explorer is designed to gather experiential knowledge of tool usage. Specifically, the Explorer simulates plausible scenarios in which the tool might be utilized, capturing tool outputs under diverse parameter configurations. To encourage **diverse exploration**, we evaluate the similarity between newly generated queries and those from earlier iterations. If the similarity is too high, the Explorer performs self-reflection to analyze the reasons for redundancy and generates more diverse queries that meet the similarity constraint. Additionally, the Rewriter incorporates feedback from both the Explorer and Analyzer in the current iteration to suggest directions for the Explorer in the next iteration. This iterative and collaborative framework facilitates comprehensive and diverse exploration of the tool.
>
> For the remark "It would be worth the authors to include in the discussion about how to generalize this for tool documents without good parameter documentation.": Excellent comment! DRAFT employs an **iterative process** that can address cases where the original tool documentation **lacks parameter details.** In such cases, the LLM may **initially fail to populate required parameters**, resulting in tool invocation errors. However, the **error messages often provide clues about the missing parameters.** These error messages then guide the LLM in subsequent iterations to generate more accurate queries. For example, as shown in the tool documentation **revision trajectory** detailed in **Appendix E**, during the first two iterations, the LLM failed to fill the required "person_id" parameter, resulting in the tool returning an error: "Invalid id: The pre-requisite id is invalid or not found." This feedback enabled the LLM to correctly populate the parameter in the third iteration, obtain the correct output, and subsequently refine the tool documentation.
>
> Thus, through this feedback-driven iterative framework, DRAFT can effectively handle incomplete tool documents and, over multiple iterations, extract the missing parameter information from tool responses to create improved documentation. **Future studies** could explore optimizing this process to **reduce the number of iterations** required when addressing such cases. For instance, **one potential direction** could involve more sophisticated processing of **tool responses**, as they are the primary source for uncovering missing parameter information in such scenarios.

---

> ### Author Response · Authors · 2024-11-19
> **Response to Reviewer JQEK : Part 2**
>
> > [W2] The exploration-revision loop essentially helps the system to collect more input-output examples that previously doesn't exist in the document. A **qualitative analysis** between examples generated by revision-only based tool like EasyTool and new examples generated by DRAFT would be helpful to understand this nuance difference.
>
> **Response:** We thank the reviewer for this valuable suggestion. EasyTool does incorporate simulated tool function guidelines into the tool documentation to assist the LLM in understanding the content. However, it **does not involve actual tool invocations to construct input-output examples** for documentation refinement. Instead, EasyTool relies solely on the original tool documentation and the LLM's interpretation to modify the documentation. As a result, its modifications are limited to the original tool documentation without integrating feedback from the LLM's interactions or the tool's actual outputs. Additionally, EasyTool **performs only a single round of revision**, whereas DRAFT adopts an iterative approach, enabling the collection of multiple rounds of input-output examples.
>
> To highlight the advantages of DRAFT in this regard, we conduct an evaluation using GPT-4o as the evaluator to assess whether examples generated by DRAFT or EasyTool exhibit higher variability in terms of parameter settings, edge-case scenarios, and less frequently used functionalities. The results are as follows:
>
> | Dataset | DRAFT | EasyTool | Equal |
> | :---: | :---: | :---: | :---: |
> | RestBench | 100 | 0 | 0 |
> | ToolBench | 100 | 0 | 0 |
>
> The results clearly demonstrate that examples generated by DRAFT are more diverse and better cover a broader range of scenarios compared to EasyTool on both benchmarks. This can be attributed to several factors:
> 1. **Iterative Refinement:** DRAFT's iterative approach allows for the generation of multiple input-output examples over successive iterations.
> 2. **Diversity-Promoting Exploration:** DRAFT employs a strategy specifically designed to encourage diverse exploration, leading to more varied and comprehensive examples.
> 3. **Feedback Integration:** EasyTool performs only a single round of revision, generating a single set of tool function guidelines without incorporating feedback from actual tool outputs.
>
> Furthermore, the experimental results in **Table 1** of our manuscript also confirm that tool documentation refined by DRAFT outperforms that refined by EasyTool in terms of downstream performance.

---

> ### Author Response · Authors · 2024-11-19
> **Response to Reviewer JQEK : Part 3**
>
> > [W3] The intro motivates different types of documentation limitations (incomplete, redundant, inaccurate), it would be good to provide **qualitative feedback** about which scenarios DRAFT is best at improving, especially comparing to prior tools.
>
> **Response:** We thank the reviewer for the insightful suggestion. We also consider exploring which specific documentation limitations
>  DRAFT is most effective at addressing. However, the benchmarks we used and real-world query/instruction scenarios often require **multiple tools** to work together to solve a single task. This makes it **challenging to disentangle the downstream performance improvements and attribute them to specific enhancements** in documentation quality. For example, if a query requires three tools to function effectively—one with incomplete documentation, another with redundant information, and the third with inaccuracies—the overall downstream improvement likely stems from combined enhancements across all three tools, making it **difficult to isolate the impact of any one type of documentation improvement.** Additionally, a **single tool documentation may exhibit multiple types** of limitations simultaneously (e.g., being both incomplete and inaccurate), making it **hard to group tools solely** by their limitations and conduct targeted experiments.
>
> However, our **human evaluation results in Table 4** provide some useful insights. Annotators compare the original documentation with the DRAFT-modified versions across three dimensions: completeness, conciseness, and accuracy. The results show that DRAFT consistently achieves the **greatest improvements in the completeness dimension** across both benchmarks. To provide a more quantitative analysis, we calculated scores based on the following rule:
> $$
> s = \delta_{\text{DRAFT win}} - \delta_{\text{Raw win}},
> $$
> where $\delta_{\text{DRAFT win}}$ represents the number of instances where the DRAFT-modified tool documentation is evaluated as better, and $\delta_{\text{Raw win}}$ represents the number of instances where the Raw tool documentation is rated as better. Note that the **score $s>0$ indicates that DRAFT-modified tool documentation is evaluated as better more frequently**, while $s<0$ indicates the opposite. The **absolute value of $s$ reflects the extent of this tendency**. The results are summarized below:
>
> | Dataset | Completeness | Conciseness | Accuracy |
> | :---: | :---: | :---: | :---: |
> | RestBench | 24 | 16 | 30 |
> | ToolBench | 64 | 52 | 56 |
>
> To ensure the reliability of these results, we also conducted the same evaluation using GPT-4o as an evaluator. The results are as follows:
> | Dataset | Completeness | Conciseness | Accuracy |
> | :---: | :---: | :---: | :---: |
> | RestBench | 84 | 12 | 45 |
> | ToolBench | 86 | 34 | 64 |
>
> Both human annotators and GPT-4o evaluations indicate that DRAFT-modified documentation is **consistently preferred** over the original documentation across all dimensions (as evidenced by positive scores in all categories), further validating the advantages of DRAFT. Meanwhile, both human annotators and GPT-4o consistently found that DRAFT-modified documentation **scored highest in the Completeness dimension, followed by Accuracy.** This suggests that DRAFT is particularly effective at making documentation more complete. Our framework leverages feedback from LLM interactions and tool responses to enrich the original documentation, thereby addressing missing information and improving completeness. While DRAFT also replaces redundant or irrelevant information with useful content and corrects inaccuracies, its core strength lies in increasing the informational richness of the documentation, making it more comprehensive.
>
> **We hope the above response will fully address your concerns about our work.** We really appreciate your insightful and constructive comments to further help us improve the quality of our manuscript. Thanks again! Please do not hesitate to let us know if you have any further questions.

---

> > ### Comment · Reviewer_JQEK · 2024-12-03
> > **Thanks for the response**
> >
> > I appreciate authors' discussion here. Practically, this approach would have most impact on scenarios where initial documentation is not well crafted (e.g., missing args, incomplete functionality), but I think these would be future work.
> >
> > I remain positive about the paper. Thanks for the contribution.

---

> > > ### Author Response · Authors · 2024-12-03
> > > **Thanks for the Positive Feedback**
> > >
> > > Thanks very much for your positive feedback and insightful comments. We will explore this issue more deeply in our future work. We appreciate your support and are grateful for your constructive feedback!
> > >
> > > Best regards,
> > >
> > > Authors

---

### Official Review · Reviewer_C1G3 · 2024-11-03

**Soundness:** 3
**Presentation:** 4
**Contribution:** 4
**Rating:** 8
**Confidence:** 4

**Summary:**

This paper highlights the importance of proper tool documentation to improve tool learning. Due to incomplete documentation or consideration of human intuition, tool documentation may not include all the information to be useful for an LLM. To resolve these issues, the paper proposed DRAFT that iteratively gathers ‘experience’, learns from it, and rewrites the documentation for future usage.

**Strengths:**

1. DRAFT significantly outperforms across two benchmarks and outperforms three baselines.
2. It also verifies the results through a small human study — showing its effectiveness in practical use cases.
3. The paper provides an in-depth analysis of the impact of iteration count on the performance as well as the improved performance for retrieval in the rewritten documentation.

**Weaknesses:**

1. According to the first contribution highlighted in the introduction, DRAFT highlights ‘inefficiencies and inaccuracies within existing tool documentation hamper the effective utilization of tools by LLMs’. However, the paper does not show any quantitative evaluation of this claim rather than explaining the problems associated with it. Hence, I think the ‘highlight’ itself may not be a novel contribution as much as the framework.
2. For ‘tool-adaptive termination mechanism’, is it possible to measure the impact of different threshold on the performance of DRAFT?

**Questions:**

1. The paper highlights how the count of iteration impacts the performance of DRAFT mentioning - ‘a decline in performance is observed after a certain number of iterations. This decline may be attributed to the introduction of redundant information as the number of iterations increases, potentially leading to overfitting’. Is it possible to measure the count of redundant information in the system? It would be nice to see some sort of hallucination error measurement on top of DRAFT.

---

> ### Author Response · Authors · 2024-11-19
> **Response to Reviewer C1G3 : Part 1**
>
> We sincerely thank the reviewer for the constructive comments and suggestions, which are very helpful for improving our paper. We are also grateful that you recognized the strengths of our paper. The revisions have been incorporated in the revised manuscript marked in blue. Please kindly find point-to-point responses below.
>
> > [W1] According to the first contribution highlighted in the introduction, DRAFT highlights 'inefficiencies and inaccuracies within existing tool documentation hamper the effective utilization of tools by LLMs'. However, the paper does not show any quantitative evaluation of this claim rather than explaining the problems associated with it. Hence, I think the 'highlight' itself may not be a novel contribution as much as the framework.
>
> **Response:** We thank the reviewer for the insightful concern. In the introduction, we include some quantitative evaluation experiments; specifically, **Figure 1(c)** presents a **pairwise comparison** between the original and modified tool documentation to illustrate which version is more understandable for LLMs. This comparison shows that the modified documentation is more effective for LLMs. The reasons provided are that the documentation refined by DRAFT is more detailed and clear, with additional features such as error handling, which **highlight some of the limitations in the original documentation.**
>
> Additionally, to further demonstrate how issues in original documentation might hinder LLMs' effective tool use, we conduct a **pointwise evaluation** where the LLM assesses whether the documentation meets the criteria of completeness, conciseness, and accuracy. The results are presented as follows:
>
> | Datasets | Yes | No |
> | --- | --- | --- |
> | RestBench-TMDB | 22.22% | 77.78% |
> | RestBench-Spotify | 17.50% | 82.50% |
> | ToolBench | 12.06% | 87.94% |
>
> It can be seen that **most original tool documentation falls short of these criteria**, providing **quantitative evidence** of existing documentation issues, which, as illustrated in Figure 1(a), can lead to challenges such as the inability to correctly select tools or fill in the necessary parameters for tool invocation.
>
> Finally, as seen in the results of our main downstream experiment in **Table 1**, the performance using DRAFT-modified documentation **surpasses that of the original documentation**, further supporting our claim that original documentation can indeed hamper the effective utilization of tools by LLMs.
>
> > [W2] For 'tool-adaptive termination mechanism', is it possible to measure the impact of different threshold on the performance of DRAFT?
>
> **Response:** We thank the reviewer for the valuable question. Intuitively, considering **extreme cases**, when the threshold is set to 0, any similarity between iterations is deemed "too similar," causing the process terminating after the first round. Conversely, when the threshold is set to 1, no early stopping occurs, as all iterations are considered dissimilar, and the process continues until reaching the maximum iteration limit of 5. This shows that the number of iterations before termination is closely related to the threshold setting.
>
> In **Section 3.3**, our analysis on iteration rounds shows that performance tends to initially improve with more iterations before eventually declining, suggesting that the threshold choice may similarly influence performance. To further investigate this effect, we **conduct experiments using various threshold values**. The results are shown as follows:
>
> | Threshold | RestBench-TMDB |  | RestBench-Spotify |  | ToolBench |  |
> | :---: | :---: | :---: | :---: | :---: | :---: | :---: |
> |  | CP% | Win% | CP% | Win% | CP% | Win% |
> | 0 | 82.00 | 67.00 | 64.91 | 73.68 | 44.00 | 71.00 |
> | 0.25 | 82.00 | 67.00 | 64.91 | 73.68 | 44.00 | 71.00 |
> | 0.5 | 83.00 | 69.00 | 68.42 | 77.19 | 47.00 | 72.00 |
> | 0.75 | 88.00 | 71.00 | 70.17 | 84.21 | 51.00 | 78.00 |
> | 1 | 80.00 | 68.00 | 59.64 | 68.42 | 46.00 | 76.00 |
>
> We observe that performance follows a similar trend as with the iteration count, **initially increasing with higher thresholds before eventually declining**. For example, when the threshold is set very low (0 or 0.25), iterations stop after the first round, resulting in lower performance. As the threshold increases, more iterations are allowed, which enhances performance. However, when the threshold is too high, iterations only terminate at the maximum limit, which leads to performance decline. While 0.75 is not necessarily the absolute optimal threshold, we observe that it produces satisfactory results within our budget constraints. Therefore, we did not experiment with additional values beyond this point.

---

> ### Author Response · Authors · 2024-11-19
> **Response to Reviewer C1G3 : Part 2**
>
> > [Q1] The paper highlights how the count of iteration impacts the performance of DRAFT mentioning - 'a decline in performance is observed after a certain number of iterations. This decline may be attributed to the introduction of redundant information as the number of iterations increases, potentially leading to overfitting'. Is it possible to measure the count of redundant information in the system? It would be nice to see some sort of hallucination error measurement on top of DRAFT.
>
> **Response:** We thank the reviewer for raising such valuable question. The redundancy we refer to here **encompasses various types of information**, such as repetitive content, irrelevant details, and hallucinated information. The concept of hallucination is indeed broad. For example, while the LLM may generate factually correct information, details that do not aid in effective tool use can also be considered hallucinatory in this context. In **Appendix Section E**, we included an example of a tool documentation **revision trajectory**. This example shows that as the number of iterations increases, the tool documentation gradually becomes **longer**. To further explore the relationship between the iteration count and documentation length, we conduct a quantitative analysis. The results are presented below:
>
> | iteration rounds | RestBench-TMDB | RestBench-Spotify | ToolBench |
> | :---: | :---: | :---: | :---: |
> | 1 | 71.92 | 66.200 | 20.9184 |
> | 2 | 92.14 | 86.075 | 28.9625 |
> | 3 | 107.31 | 99.925 | 31.6007 |
> | 4 | 115.03 | 109.575 | 35.5295 |
> | 5 | 128.72 | 105.700 | 38.6082 |
>
> Across three datasets, we observe a trend where documentation length increases with each iteration. Although increased length does not necessarily imply added irrelevant information, previous studies, such as Nelson F. Liu (2023) [1], suggest that **LLMs may struggle to effectively utilize information from longer texts**. Therefore, as tool documentation lengthens, the LLM's ability to use the tool effectively may be impaired.
>
> To examine this effect further, we measure the **ratio of information entropy gain to the increase in document length** between successive iterations, defining this as the effective length increase. The results are as follows:
>
> | iteration rounds | RestBench-TMDB | RestBench-Spotify | ToolBench |
> | :---: | :---: | :---: | :---: |
> | 1->2 | 1.30 | 1.92 | 5.55 |
> | 2->3 | 1.19 | 0.97 | 4.08 |
> | 3->4 | 1.17 | 0.85 | 3.66 |
> | 4->5 | 0.67 | 1.53 | 3.41 |
>
> We observe only one outlier, which is caused by a decrease in length. In all other cases, the results shows that the effective length increase **diminishes with more iterations**, indicating that useful information gains decrease over time. This trend provides a **quantitative measure** of redundancy in later iterations.
>
> [1] Liu NF et.al Lost in the middle: How language models use long contexts, TACL 2024
>
> **We hope the above response will fully address your concerns about our work.** We really appreciate your insightful and constructive comments to further help us improve the quality of our manuscript. Thanks again! Please do not hesitate to let us know if you have any further questions.

---

> > ### Comment · Reviewer_C1G3 · 2024-12-02
> > **Response to Authors**
> >
> > Thanks to the authors for their detailed answers. This answers my question and I do not have any further critical feedback or concern. I would encourage the authors to include the new results in the ablation study / appendix of the paper. It might be nice to include the quantitative values in the main contribution making it clear how the inefficiencies and inaccuracies impact the performance.

---

> ### Author Response · Authors · 2024-12-03
> **Thanks for the Positive Feedback**
>
> Thanks very much for your positive feedback and detailed comments. We will put our responses into our final version. Thanks again for your valuable suggestions!
>
> Best regards,
>
> Authors

---

### Author Response · Authors · 2024-11-24
**Revised pdf and summary**

Dear Reviewers,

Thanks for your hard work, your suggestions really help us to improve our paper. We revised our paper according to your suggestions (**revised parts are marked as blue**) and **re-upload our modified pdf.**

We summarize the changes and addressed concerns as follows:

- We **correct all typos** in the paper to enhance its overall quality and clarity.
- In **Section 2**, we add more citations and provide detailed explanations in our responses to address key concerns, including:
    - Why contributions such as diversity and termination mechanisms are necessary.
    - The effectiveness of self-reflection.
    - The varying difficulty of mastering different tools.
    - Why combining similarity score with BLEU score is effective for measuring the degree of change.
- In **Section 3**, we improve the **implementation details** of the experiments.
- We add a more detailed description of **human evaluation** in **Appendix A**.
- We conduct additional quantitative experiments to demonstrate:
    - The issues in original tool documentation that hinder LLMs' effective use of tools.
    - The increase in redundant information over iterations.
    - The impact of different thresholds on DRAFT's performance.
- We further include quantitative experiments showing:
    - DRAFT-generated examples are more diverse compared to EasyTool.
    - DRAFT is particularly effective at improving the completeness of tool documentation.

Finally, we emphasize that our paper primarily addresses a critical yet often overlooked issue: human-written tool documentation frequently hinders the effective utilization of tools by LLMs. To tackle this, we propose a novel framework, DRAFT, designed to dynamically adjust and optimize tool documentation based on interaction feedback between LLMs and external tools. This approach significantly enhances the overall tool-using capabilities of LLMs. We kindly ask you to consider both the **contributions of our work and its potential real-world applications** for the tool learning community.

If you have any questions, please be free to ask them before the deadline (**Nov. 26**), we will answer them as soon as possible.

Best,

Authors

---

### Meta-Review · Area_Chair_SLpV · 2024-12-19

**Metareview:**

The paper highlights the importance of proper documentation for tools in the context of LLM tool usage. They propose a method to improve this documentation, and show its effectiveness. The paper provides an important observation along with a novel solution for the issue. The reviews appreciated the depth of the analysis, including not just the bottom-line results but motivation for chosen made. The positioning against related work is agreed by the reviewers to be well-written and thorough, and the empirical evidence of the advantages provided by the technique is solid and convincing.

One key issue, raised by CrZf, was a need for a better comparison with a previous work (DSPy), but this was mitigated by the authors’ response. I urge the authors to make this issue clear in the final version of the paper. Concluding, the paper provides a novel solution for an important problem along with convincing empirical evidence of its effectiveness. It would be a great addition to ICLR.

**Additional Comments On Reviewer Discussion:**

see meta review

---

### Decision · Program_Chairs · 2025-01-22

Accept (Oral)